



# Upside-down fluxes Down Under: $CO_2$ net sink in winter and
# net source in summer in a temperate evergreen broadleaf forest
Alexandre A. Renchon[1], Anne Griebel[1], Christopher A. Williams[2], Belinda Medlyn[1], Remko A.
Duursma[1], Craig VM Barton[1], Chelsea Maier[1], Matthias M. Boer[1], Peter Isaac[3], David Tissue[1], Victor
Resco de Dios[4], Elise Pendall[1]
*[1]Hawkesbury Institute for the Environment, Western Sydney University, Penrith, NSW, Australia.*
*[2]Clark University, Graduate School of Geography, Worcester, Massachusetts 01610, USA.*
*[3]CSIRO Oceans & Atmosphere Flagship, Yarralumla, ACT, 2600, Australia.*
*[4]Department of Crop and Forest Sciences and Agrotecnio Center, University of Lleida, E-25198*
*Lleida, Spain.*
## Abstract
Predicting the seasonal dynamics of ecosystem carbon fluxes is challenging in broadleaved evergreen
forests because of their moderate climates and subtle changes in canopy phenology. We assessed the
climatic and biotic drivers of the seasonality of net ecosystem-atmosphere $CO_2$ exchange (NEE) of a
eucalyptus-dominated forest near Sydney, Australia, using the eddy covariance method. The climate
is characterized by a mean annual precipitation of 800 mm and a mean annual temperature of 18 °C,
hot summers and mild winters, with highly variable precipitation. In the three-year study, the
ecosystem was a small sink in 2014 (54 g C $m^{-2}$ $y^{-1}$), a stronger sink in 2015 (183 g C $m^{-2}$ $y^{-1}$) and
even stronger sink in 2016 (337 g C $m^{-2}$ $y^{-1}$), but these variations were not related to precipitation.
Daily net C uptake was always detected during the cooler, drier winter months (June through August),
while net C loss occurred during the warmer, wetter summer months (December through February).
Gross primary productivity (GPP) seasonality was low, despite longer days with higher light intensity
in summer, because vapour pressure deficit (D) and air temperature ($T_a$) restricted surface
conductance during summer while winter temperatures were still high enough to support
photosynthesis. Maximum GPP during ideal environmental conditions was correlated with canopy
leaf area index (LAI) ($r^2 = 0.24$), which increased rapidly after mid-summer rainfall events.
Ecosystem respiration (ER) was highest during summer in wet soils and lowest during winter months.
ER had larger seasonal amplitude compared to GPP, and therefore drove the seasonal variation of
NEE. Because summer carbon uptake may become increasingly limited by atmospheric drought and
high temperature, and ecosystem respiration could be enhanced by rising temperature, our results
suggest the potential for large-scale seasonal shifts in NEE in sclerophyll vegetation under climate
change.
*Keywords*: net ecosystem exchange, seasonal variability, atmospheric drought, canopy phenology





## 1. Introduction

Forests and semi-arid biomes are responsible for the majority of global carbon storage by terrestrial ecosystems (Dixon et al., 1994;Schimel et al., 2001;Pan et al., 2011;Poulter et al., 2014). Photosynthesis and respiration by these biomes strongly influence the seasonal cycle of atmospheric $CO_2$ (Keeling et al., 2001;Baldocchi et al., 2016). Continuous measurements of land-atmosphere exchanges of carbon, energy and water provide insights into the seasonality of forest ecosystem processes, which are driven by the interactions of climate, plant physiology and forest composition and structure (Xia et al., 2015). Net ecosystem exchange (NEE) seasonality is relatively well understood in cool-temperate ecosystems; deciduous trees can only photosynthesize when they have leaves and NEE dynamics are thus principally influenced by the phenology of canopy processes. NEE of deciduous forests thus has a more pronounced seasonality than that of evergreen conifer forests at similar latitudes (Novick et al., 2015). For high-latitude evergreen conifer forests, NEE seasonality is strongly limited by cold temperature limitation of photosynthesis (Kolari et al., 2007) and respiration. In contrast, seasonality of NEE in evergreen broadleaf forests, typically occurring in warm-temperate and tropical regions, is much less well understood (Wu et al., 2016;Restrepo-Coupe et al., 2017).

The seasonality of gross primary productivity (GPP) in evergreen broadleaf forests may be driven by climate (e.g. dry/wet seasons) and/or by canopy dynamics (Wu et al., 2016). In tropical evergreen forests, air temperature and day length are similar seasonally, but precipitation seasonality can be strong, with higher radiation and temperature (1 or 2 °C higher) in the dry season (Trenberth, 1983;Windsor, 1990). Counter-intuitively, GPP can be higher during the dry season, as cloud cover may limit productivity in the wet season (Graham et al., 2003;Saleska et al., 2003;Hutyra et al., 2007). Canopy dynamics can be an important determinant of GPP seasonality in evergreen broadleaf forests; although leaves are present in the canopy year-round in evergreen canopies, LAI may show considerable temporal variability seasonally as new leaves are produced and old leaves die, especially during leaf flush and senescence periods (Duursma et al., 2016;Wu et al., 2016). The leaf light use efficiency and water use efficiency may both vary as leaves age: young leaves and old leaves are less efficient than mature leaves, reflecting changes in photosynthetic capacity (Wilson et al., 2001;Wu et al., 2016). The timing of leaf flush and senescence can depend on the environment and on species; environmental stress, such as drought, can induce the process of senescence (Munné-Bosch and Alegre, 2004;Lim et al., 2007).

In temperate evergreen broadleaved forests, such as eucalypt-dominated sclerophyll vegetation in Australia, precipitation can be seasonal or aseasonal; furthermore, day length and temperature vary significantly between winter and summer. GPP can be limited by frost during winter and by drought during summer. Atmospheric drought indicated by high vapor pressure deficit (D), and soil drought have different impacts on GPP, but they can interact to impact surface conductance ($G_s$) (Medlyn et al., 2011;Novick et al., 2016). In Australia's temperate eucalypt forests, canopy rejuvenation takes place in summer and is linked to heavy rainfall events (Duursma et al., 2016). However, since leaf flushing and shedding occur simultaneously in eucalypt canopies (Pook, 1984;Duursma et al., 2016), the overall canopy volume can remain stable while the distribution of canopy volume changes with height (Griebel et al., 2015). Eucalypt forests in southeast Australia have been found to act as carbon sinks all year long, with greater uptake in summer (van Gorsel et al., 2013;Hinko-Najera et al., 2017). Although canopy characteristics are key to understanding ecosystem fluxes, their dynamics in Australian ecosystems can be particularly challenging to detect using standard vegetation indices (Moore et al., 2016). Nevertheless, the normalized difference vegetation index (NDVI) has successfully explained variability in photosynthetic capacity in Mediterranean, mulga and savanna ecosystems (Restrepo-Coupe et al., 2016).



The environmental and biotic controls on the seasonal dynamics of ecosystem fluxes in broadleaved
evergreen forests are still poorly understood. Our objective was to determine the seasonality of
ecosystem $CO_2$ and $H_2O$ fluxes in a dry sclerophyll Eucalyptus forest; we evaluated the role of
environmental drivers (PAR, $T_a$, SWC and D) and canopy dynamics (as measured with NDVI, LAI,
litter fall and leaf age) in regulating the seasonal patterns of net ecosystem exchange (NEE), gross
primary productivity (GPP), ecosystem respiration (ER), evapotranspiration (ET) and surface
conductance ($G_s$) in an evergreen forest near Sydney, Australia. We also compared leaf-level to
ecosystem-level water and carbon exchange in response to drivers, in order to gain confidence in our
results and gain insights about the emergent properties from leaf to ecosystem scale. We hypothesised
that canopy phenology (LAI and leaf age) explains temporal variation in photosynthetic capacity (PC)
and $G_s$. We anticipated that the ecosystem would be a carbon sink all year long.



## 2. Material and methods

*2.1 Site description*

The field site is the Cumberland Plain forest SuperSite (de Dios et al., 2015) of the Australian Terrestrial Ecosystem Research Network (http://www.tern.org.au/), located 50 km west of Sydney, Australia, at 19 m elevation, on a nearly flat floodplain of the Nepean-Hawkesbury River (latitude -33.61320; longitude 150.72446). Mean mid-afternoon (3 pm) temperature is 18 °C (max. 28.5 °C in January and min. 16.5 °C in July) and average precipitation is 801 mm year$^{-1}$ (mean monthly max. is 96 mm in January, and min. is 42 mm in September). The soil is classified as a Kandosol and consists of a fine sandy loam A horizon (0-8 cm) over clay to clay loam subsoil (8-40 cm), with pH of 5 to 6 and up to 5% organic C in the top 10 cm (Karan et al., 2016). The flux tower is in a dry sclerophyll forest, with 140 Mg C ha$^{-1}$ aboveground biomass and stand density of ~500 trees ha$^{-1}$. The predominant canopy tree species are *Eucalyptus moluccana* and *E. fibrosa*, which grow to an average canopy height of ~25 m and host a large population of mistletoe (*Amyema miquelii*). In addition, the mid-canopy (5-12 m) is dominated by *Melaleuca decora*, the understory is dominated by *Bursaria spinosa* with various shrubs, forbs, grasses and ferns present in lower abundance.

*2.2 Environmental measurements*

Air temperature ($T_a$) and relative humidity (RH) were measured using HMP45C (Vaisala, Vantaa, Finland) sensors at 7 m and 29 m heights. Vapour pressure deficit (D) was estimated from $T_a$ and RH. Photosynthetically active radiation above the canopy (PAR, W m$^{-2}$) was measured using an LI190SB (Licor Inc., Lincoln NE, USA), and incoming and outgoing short and longwave radiation were measured using a CNR1 radiometer (Kipp & Zonen, Delft, Netherlands). Ancillary data were logged on CR1000 or CR3000 dataloggers (Campbell Scientific, Logan UT, USA) at 30 min intervals. Mixing ratios of $CO_2$ in air were also measured at 0.5 m, 1 m, 2 m, 3.5 m, 7 m, 12 m, 20 m, and 29 m above the soil surface using a LI840A Gas Analyzer (Licor Inc., Lincoln NE, USA); data from each height were logged on a CR1000 datalogger once every 30 minutes (1 minute air sampling per height).

Ground heat flux and soil moisture were averaged between two locations to represent the variable shading in the tower footprint. One location had a HFP01 heat flux plate and the other has a self-calibrating heat flux plate (HFP01SC) (Hukseflux, XJ Delft, Netherlands) installed at 8 cm below the soil surface. The heat flux plates were paired with a CS616 water content reflectometer (Campbell Scientific, Logan UT) installed horizontally at 5 cm below the soil surface and a TCAV averaging thermocouple (Campbell Scientific, Logan UT) installed with thermocouples at 2 cm and 6 cm below the soil surface for each pair. A CS616 installed vertically measured average soil water content from 7 to 37 cm (CS616). Rainfall was measured at an open area with a tipping bucket 2 km away from the study site.

*2.3 Net ecosystem exchange*

Continuous land-atmosphere exchange of $CO_2$ mass (net ecosystem exchange, NEE) was quantified from direct measurements of the different components of the theoretical mass balance of $CO_2$ in a control volume:

$$NEE = F_{CT} + F_{CS} + (F_{CAH} + F_{CAV}) \qquad (1)$$

Where $F_{CT}$ is the vertical turbulent exchange flux, $F_{CS}$ is the change in storage flux, and $F_{CAH}$ and $F_{CAV}$ are the horizontal and vertical advection fluxes –assumed negligible when atmospheric turbulence is sufficient (Baldocchi et al., 1988;Aubinet et al., 2012). We used change-point detection of the friction



velocity (u*) threshold (Barr et al., 2013) to determine the turbulence threshold above which NEE (the
sum of $F_C$ and $S_C$) is independent of u*. However, we found no clear dependence of NEE on u* hence
no clear threshold (Figure S1), so we used a threshold of 0.2 m s$^{-1}$ to be conservative.
The calculation of each term, and the assumptions required for them to be representative of each half-
hour flux are detailed below.

### 140    *2.4 Vertical turbulent flux ($F_{CT}$)*

The vertical turbulent fluxes of $CO_2$ ($F_{CT}$, µmol m$^{-2}$ s$^{-1}$) and water ($F_{WT}$, mmol m$^{-2}$ s$^{-1}$) were measured
using the eddy-covariance method (Baldocchi et al., 1988). Concentration (c) of $CO_2$ or water vapor
(open-path IRGA (LI-7500A, Licor Inc., Lincoln NE, USA)) and vertical wind speed (w) (CSAT 3D
sonic anemometer (Campbell Scientific, Logan UT, USA)) were measured at 10Hz frequency at 29 m
above the ground, and logged on a CR-3000 datalogger (Campbell Scientific, Logan UT, USA). $F_{CT}$
and $F_{WT}$ are calculated as the average 30 minute covariance of c and w:

$$F_{CT} \text{ or } F_{WT} = \overline{w'c'} \tag{2}$$

Vertical turbulent fluxes were calculated from the 10 Hz data, using Eddy-Pro© software. The
calculation allowed for up to 10% of missing 10 Hz data. Axis rotation for tilt correction used the
double rotation method (Wilczak et al., 2001). Time lags were compensated using covariance
maximization, within a window of plausible time lags (Fan et al., 1990). We applied the block
averaging method for de-trending (Gash and Culf, 1996). Density fluctuations in the air volume were
corrected using the WPL terms (Webb et al., 1980). Statistical tests for raw data screening followed
(Vickers and Mahrt, 1997), including spike count/removal, amplitude resolution, drop-outs, absolute
limits and skewness and kurtosis tests. Low and high frequency spectral correction followed
(Moncrieff et al., 2004), and (Moncrieff et al., 1997). Each half-hourly flux was associated with a
quality flag (0: good quality, 1: keep for integrations, discard for empirical relationships, 2: remove
from the data); these flags accounted for stationarity tests and turbulence development tests which are
required for good turbulent flux measurements (Foken et al., 2004). In our 3-year record, 51% of $F_{CT}$
fluxes had a flag of 0, 32% had a flag of 1 and 17% had a flag of 2.

### 160    *2.5 Storage flux ($F_{CS}$)*

The change in storage flux ($F_{CS}$, µmol m$^{-2}$ s$^{-1}$) was measured using a $CO_2$ profiler system, such that
change of storage flux timestamp was the same as the turbulent flux timestamp. The change in storage
flux was calculated as (Aubinet et al., 2001):

$$F_{CS} = \frac{P_a}{R \, T_a} \int_0^h \frac{dC(z)}{dt} dz \tag{3}$$

Where $P_a$ is the atmospheric pressure ($P_a$), $T_a$ is the temperature (K), R is the molar gas constant, and
C(z) is $CO_2$ (ppm) at the height z. As we only measure a limited number of heights, this equation
becomes, in practice:

$$F_{CS} = \left(\frac{\Delta C}{\Delta t}\right)_{k=1} \times z_{k=1} + \sum_{k=2}^{n} \left\{ \left[ \left(\frac{\Delta C}{\Delta t}\right)_k + \left(\frac{\Delta C}{\Delta t}\right)_{k-1} \right] \times \frac{z_k - z_{k-1}}{2} \right\} \tag{4}$$

Where k [1 to 8] represents each inlet height. $T_a$ was linearly interpolated from HMP at 29 m and 7 m.
We flagged and replaced the storage flux with a one-point approximation during profiler outages
(25% of the 3-year record), using the change in $CO_2$ at 29 m height over 30 minutes as derived in
EddyPro (Aubinet et al., 2001). These data were not used for empirical relationships, but kept for




annual sum calculations. Storage flux of water vapour was assumed to be negligible. For visualisation
of the diurnal course of storage flux and turbulent flux, see Figure S2.
*2.6 Gap-filling of environmental variables and NEE separation into gross fluxes*
The OzFluxQC processing software, based on SOLO neural network, was used for gap-filling
climatic variables and fluxes and for partitioning the NEE into gross primary productivity (GPP) and
ecosystem respiration (ER) (Hsu et al., 2002;Isaac et al., 2017), using data with QC flags of 0 and 1
(Foken et al., 2004). In brief, gaps were filled following the hierarchy of using variables provided
from 1) automatic weather stations from the closest weather station, 2) numerical weather prediction
model outputs (ACCESS regional, 12.5 km grid size provided by the Bureau of Meteorology) and
lastly 3) monthly mean values from the site-specific climatology. In a next step the continuous climate
variables were used to fill all fluxes by utilizing the SOLO neural network with 25 nodes and 500
iterations on monthly windows. We selected the drivers based on the highest $r^2$, which identified net
radiation ($F_n$), specific heat density (SHD), air temperature ($T_a$) and wind speed ($w_s$) for latent heat
flux; $F_n$, $T_a$ and $w_s$ for sensible heat flux and D, $T_a$, shortwave incoming radiation ($F_{sd}$) for NEE. In
addition, all nocturnal observational data (at night, we assume GPP = 0 so NEE = ER) that passed all
quality control checks and the u*-filter were modelled using $T_s$, $T_a$ and SWC as drivers. Lastly, this
gapfilled data (gap-filled ER from nighttime NEE) was used to infer GPP as the result of NEE - ER.
*2.7 Flux footprint*
We analysed which turbulent flux was out of the footprint according to (Kljun et al., 2004), using a
criterion that at least 75% of the turbulent flux ($F_{CT}$) should come from within the forest area. We
used Nearmap high resolution aerial imagery to determine the extent of the forest ecosystem
surrounding the tower. We found that, after u* filtering, $CO_2$ turbulent fluxes ($F_{CT}$) originated from
the footprint of interest. We assumed that the ecosystem within the footprint was homogeneous for the
purpose of this study.
*2.8 Energy balance*
We evaluated the energy balance closure with the ratio of available energy (net radiation ($R_n$) – soil
heat flux (G)) to the sum of turbulent heat fluxes (latent heat flux (L) + sensible heat flux (H)). On a
daily basis, the energy balance closure was 70% (Figure S3), consistent with the well-known and
common issue of a lack of closure (Wilson et al., 2002;Foken et al., 2006;Foken, 2008).
*2.9 Surface conductance*
Surface conductance ($G_s$) was derived by inverting the Penman-Monteith equation (Monteith, 1965):

$$G_s = \frac{\gamma \, L \, g_a}{\varepsilon \, R_n + \rho \, C_p \, D \, g_a - L \, (\varepsilon + \gamma)} \qquad (6)$$

Where $\gamma$ is the temperature dependent psychrometric constant (kPa $K^{-1}$), L is the latent heat flux (W
$m^{-2}$), ε is the temperature dependent slope of the saturation-vapor pressure curve (kPa $K^{-1}$), $R_n$ is net
radiation (W $m^{-2}$), ρ is the dry air density (kg $m^{-3}$), D is vapor pressure deficit (kPa), $C_p$ is the specific
heat of air (J $kg^{-1}$ $K^{-1}$), and $g_a$ is the bulk aerodynamic conductance, formulated as an empirical
relation of wind speed ($w_s$, m $s^{-1}$) and friction velocity (u*, m $s^{-1}$) (Thom, 1972):

$$g_a = \frac{1}{\frac{U}{u^{*2}} + 6.2 \, u^{*0.67}} \qquad (7)$$

In the analysis for $G_s$, we were interested in transpiration (T) rather than evaporation (E), so we
excluded data if precipitation exceeded 1 mm in the past 2 days, 0.5 mm in the past 24 hours, and 0.2



mm in the past 12 hours (Knauer et al., 2015). We assumed that evaporation (E) is negligible using
these criteria (Knauer et al., 2017), which excluded 40% of the data.
*2.10 Potential evapotranspiration*
Potential evapotranspiration rate (PET) was derived using Penman-Monteith equation (Monteith,
213   1965):

$$PET = \frac{\varepsilon R_n + C_p\, \rho\, G_a\, D}{\gamma\, [\varepsilon + \gamma(1 + \frac{G_a}{G_{s,max}})]} \tag{8}$$

where $G_{s,max}$ is the well-watered reference surface conductance, calculated as the average of $G_s$ at the
study site when soil moisture exceeds the 75% quantile and D is above 0.9 and below 1.1 kPa (Novick
et al. 2016).
*2.11 Dynamics of canopy phenology (leaf area index, litter and leaf production) and*
*photosynthetic capacity*
We evaluated the dynamics of canopy leaf area index (LAI) by measuring canopy light transmittance
with three under-canopy PAR sensors and one above canopy PAR sensor LI190SB (Licor Inc.,
Lincoln NE, USA) following the methods presented in (Duursma et al., 2016). Although we use the
term LAI, this estimate does include non-leaf surface area (stems, branches). We collected litterfall
(Lf, g m$^2$ month$^{-1}$) in the tower footprint approximately once per month, from nine litter traps (0.14 m$^-$
$^2$ ground area) located near the understory PAR sensors. We estimated specific leaf area (SLA) of
Eucalyptus and mistletoe leaves by sampling approximately 50 fresh leaves of each, in June 2017
(SLA = 56.4 cm$^2$ g$^{-1}$ for eucalyptus, 40.3 cm$^2$ g$^{-1}$ for mistletoe). For each month, we partitioned the
litter into Eucalyptus leaves, mistletoe leaves, and other (mostly woody) components. We used this
SLA to estimate leaf litter production ($L_p$) in m$^2$ m$^{-2}$ month$^{-1}$ of eucalyptus, mistletoe, and total as the
sum of both. Then, we estimated leaf growth ($L_g$, m$^2$ month$^{-2}$) as the sum of the net change in LAI
($\Delta L$) and $L_p$. Photosynthetic capacity (PC ) is defined as median GPP when PAR is 800-1200 W m$^{-2}$
and D is 1.0 to 1.5 kPa.
*2.12 Analysis of light-response of NEE*
We evaluated the light response of NEE using a saturating exponential function (Eq. 5) to test whether
parameters varied between seasons (Mitscherlich, 1909;Aubinet et al., 2001;Lindroth et al., 2008).

$$NEE = -(NEE_{sat} + R_d)\left(1 - \exp\left[\frac{-\alpha\, PAR}{NEE_{sat} + R_d}\right]\right) + R_d \tag{5}$$

where the parameter $R_d$ is the intercept, or NEE in the absence of light, often called dark respiration;
$NEE_{sat}$ is NEE at light saturation and $\alpha$ is the initial slope of the curve, expressed in µmol CO$_2$ µmol
photon$^{-1}$ and representing light use efficiency when photosynthetically active radiation (PAR) is close
to 0. We only used daytime quality checked NEE data to fit the model (qc = 0; (Foken et al., 2004),
LI-7500 signal strength = max, all inlets of profiler system data available and u* > 0.2 m s$^{-1}$), see
Figure S4.
*2.13 Leaf gas exchange spot measurements*
Spot measurement of leaf-level net photosynthesis at light saturation $A_{max}$ (PAR ~ 1800 W m$^{-2}$),
transpiration T, stomatal conductance $g_s$, and D were measured at 1.5 km for the flux tower site, see
(Gimeno et al., 2016).



## 3. Results

*3.1 Seasonality of environmental drivers and leaf area index*

The monthly average of daily maximum of air temperature was 16.3 °C during the coldest month (July 2015), and the lowest monthly average of daily maximum PAR was 878 µmol m$^{-2}$ s$^{-1}$ in the winter (June 2015; Figure 1c). Although less rainfall occurred during winter months compared to summer months, precipitation occurred throughout the year (Figure 1b). Soil volumetric water content in the shallow (0-8 cm) layer was about 10% except immediately following rain events (Figure 1b). In contrast, soil water content in the clay layer (8 -38cm) remained above 30% for the duration of the study (data not shown). Average of daily maximum of air temperature ranged from 16.3 °C in July 2015 to 31 °C in December 2016; Average of daily maximum D ranged from 0.9 kPa in June 2015 to 3 kPa in December 2016 (Figure 1c). For visualisation of seasonal and diurnal trend of radiation, air temperature, D and SWC, see supplement Figure S5.

Canopy leaf area index varied between 0.7 (in December 2014) and 1.1 m$^2$ m$^{-2}$ (in February 2015) (Figure 1d). LAI followed a distinct pattern: it peaked in late summer (around January), and then continuously decreased until the new leaves emerged the following year. Litter production was concurrent with leaf growth and also peaked in summer, before and during the leaf flush, and was lower in winter (Figure 1d).

*3.2 Seasonality of carbon and water fluxes*

Contrary to expectations, the ecosystem was always a sink for carbon in winter (-127 g C m$^{-2}$ in 2014, -135 g C m$^{-2}$ in 2015 and -99 g C m$^{-2}$ in 2016), and usually a carbon source or close to neutral in summer (+97 g C m$^{-2}$ in 2014, +31 g C m$^{-2}$ in 2015 and -15 g C m$^{-2}$ in 2016) (Table 1). Summer GPP was higher (-460 ± 112 g C m$^{-2}$) compared to winter GPP (-291 ± 28 g C m$^{-2}$) (Table 1), that is a difference of ~ 169 g C m$^{-2}$. However, summer ER was much higher (497 ± 57 g C m$^{-2}$) compared to winter ER (171 ± 26 g C m$^{-2}$) (Table 1), a difference of ~ 326 g C m$^{-2}$. The summer vs. winter ER difference was close to double the GPP difference; thus, ER had a relatively larger effect over the seasonality of NEE.

*3.3 Diurnal trend of CO$_2$ flux and drivers in winter and summer*

The diurnal pattern of NEE in clear-sky conditions differed between summer and winter (Figure 2). Relatively speaking, diurnal NEE was more symmetric in the winter than in summer. That is, morning and afternoon NEE pattern resembled a mirror image and total integrated morning NEE was similar to integrated afternoon NEE during the winter, but strong hysteresis occurred in the summer (Figure 2). This pattern also translated into hysteresis in the NEE light response curve in summer, but not in winter (Figure 3).

*3.4 Analysis of NEE light response curve*

The parameters of the NEE light response in summer and winter are shown in Figure 4 (see methods, Eq. 5). The initial slope of NEE with light (α) showed no clear dependence on T$_{soil}$ in winter but exhibited sensitivity during summer, dropping precipitously at soil temperature above 23 °C (Figure 4a). α increased with SWC in winter and summer by a factor of 2 (Figure 4b). In both winter and summer α decreased with D (D > 1 kPa) and in a similar fashion, approaching to a saturating value of 0.01 (µmol µmol$^{-1}$) at a D of about 2 kPa (Figure 4c). The fitted NEE at saturating light (NEE$_{sat}$) was not related to T$_{soil}$ in winter but decreased with increasing T$_{soil}$ in summer (Figure 4d). NEE$_{sat}$ was higher in winter than in summer for a given SWC. The relationship with D was more complicated, tending to increase with D in winter, but decreasing with increased D in summer, dropping from 9 to 3 (µmol m$^{-2}$ s$^{-1}$) as D increased from 1 to 4 kPa. R$_d$ was significantly higher in summer than winter



across all conditions of $T_{soil}$, SWC and D (Figure 4g, h, i). $R_d$ increased with $T_{soil}$ in winter and less so
in summer. $R_d$ increased with SWC in dry condition in winter and plateaued at 11%; $R_d$ was more
sensitive to SWC in summer, doubling from a rate of ~ 4 in dry soils to ~ 8 µmol m$^{-2}$ s$^{-1}$ in wet soils.
*3.5 Atmospheric and soil drought control on GPP, ET, $G_s$ and WUE*
We evaluated the effect of soil water content (SWC at 0-8 cm depth) and vapour pressure deficit (D)
on GPP, ET, water use efficiency (WUE) and canopy conductance ($G_s$) under high radiation ("PAR-
saturared"; PAR > 1000 W m$^{-2}$), after filtering periods following rain events in order to minimise the
contribution of evaporation to ET (see Methods) (Figure 5). In summer, PAR-saturated GPP
decreased above D ~ 1.3 kPa, but in winter, GPP did not vary with D. In summer and in winter, GPP
increased with SWC (Figure 5a). This is consistent with Figure 4, where $R_d$ and $NEE_{sat}$ both increased
with SWC. In summer, PAR-saturated ET increased with D up to ~1.3 kPa, above which it reached a
plateau. In winter, ET kept increasing with D, as D rarely exceeded 2 kPa. In both seasons, ET
increased with SWC (Figure 5b). Surface conductance decreased with D and SWC especially in
summer, indicating strong stomatal regulation (Figure 5b). Water use efficiency (WUE) decreased
with increasing D in summer and in winter, because ET increased but GPP declined (Figure 5c).
We compared these ecosystem-scale results to the equivalent at the leaf-scale, which are net
photosynthesis at light saturation $A_{max}$ (PAR ~ 1800 W m$^{-2}$), leaf transpiration T, leaf water use
efficiency and stomatal conductance $g_s$. These leaf level measurements are expressed on a leaf-area
basis, as opposite to ground area for ecosystem scale. We observed that $A_{max}$, T and $g_s$ were more
sensitive to D than corresponding ecosystem-scale responses. $A_{max}$ was much higher than $GPP_{max}$ at D
~ 1 kPa, while $g_s$ was comparable in magnitude to $G_s$ in the same condition. Leaf transpiration peaked
around D = 1.2 kPa, while ET plateaued. Leaf water use efficiency was overall higher than ecosystem
WUE.
*3.6 Canopy phenology control on GPP*
Monthly average of photosynthetic capacity (PC) varied by a factor of 2.7 across the study period,
ranging from 6.4 µmol m$^{-2}$ s$^{-1}$ before the leaf flush in November 2015 to 19.4 µmol m$^{-2}$ s$^{-1}$ after the
leaf flush occurred in March 2016. We expected that PC could be predicted by LAI and/or NDVI and
$G_s$. Leaf area index (LAI) and photosynthetic capacity (PC) were significantly correlated; the slope
was significantly different from zero (p = 0.02, PC = 10.9 LAI + 1.8, Figure 6). By contrast, NDVI
was not significantly correlated with PC (Figure 6). $G_{s,max}$ was significantly correlated with PC (p =
0.05, r$^2$ = 0.14, PC = 520 $G_{s,max}$ + 8.6) and LAI (p = 0.006, r$^2$ = 0.25, $G_{s,max}$ = 0.01 LAI - 1.8e-3) and
with NDVI (p = 0.003, r$^2$ = 0.24, $G_{s,max}$ = 0.015 NDVI - 4.4e-3).





## 4. Discussion

We measured three consecutive years of carbon and energy fluxes in a native evergreen broadleaf
Eucalyptus forest, including canopy dynamics and environmental drivers (photosynthetically active
radiation, air and soil temperature, precipitation, soil water content, and atmospheric demand). We
hypothesised that the Cumberland Plain forest would be a carbon sink all year-round, similar to other
eucalypt forests (Keith et al., 2012;Beringer et al., 2016;Hinko-Najera et al., 2017). We also
hypothesised higher net carbon uptake during summer, due to warmer temperatures, higher light and
longer day length contributing to higher photosynthesis, compared to winter. However, the
Cumberland Plain forest was a net source of carbon during summer, and a net sink of carbon during
winter.

The seasonal pattern of NEE was driven mostly by ER, as the seasonal amplitude of ER was larger
than the seasonal amplitude of GPP. The seasonality of ER may be explained by the positive effects
of higher temperatures on the rates of autotrophic respiration (Tjoelker et al., 2001), and on the
activity of microbes to increase soil organic matter decomposition (Lloyd and Taylor, 1994); soil
moisture remained high enough to rarely limit decomposition, especially in the subsoil. The relatively
low seasonality of GPP may be partly explained by lower photosynthetic capacity in early summer
(before January) when LAI is at its lowest, and the leaves have reached maximum age because new
leaves have not yet emerged. The ER-driven seasonality of NEE is in sharp contrast with cold
temperate forests where GPP drives the seasonality of NEE. ER-driven NEE seasonality was also
observed in an Asian tropical rain forest, as ER was higher than GPP in the rainy season leading to net
ecosystem carbon loss, while in the dry season, ecosystem carbon uptake was positive (Zhang et al.,
2010). This pattern was also observed in an Amazon tropical forest (Saleska et al., 2003).

Diurnal hysteresis of NEE in summer was associated with strong stomatal regulation, induced by high
atmospheric demand and high air temperature (Duursma et al., 2014), limiting photosynthesis during
the afternoon of warm months (see Figure S6). In winter, low D and moderately warm daytime air
temperatures and high PAR were sufficient to maintain high photosynthesis rates throughout most of
the day. Two possible explanations of the diurnal hysteresis of NEE in summer are (1) ER is greater
in the afternoon compared to morning or (2) GPP is lower in the afternoon compared to morning.
Explanation (1) is plausible, as temperature drives autotrophic and heterotrophic respiration; however,
it is unlikely as the hysteresis only happened in summer, and not in winter. Explanation (2) could arise
from lower afternoon stomatal conductance or lower photosynthetic capacity, or a combination of
both or even circadian regulation (de Dios et al., 2015;Jones et al., 1998). These diurnal patterns of
NEE, GPP and ER play a strong role in regulating the seasonal carbon cycling dynamics in this
ecosystem.

We observed comparable light saturated leaf|ecosystem $A_{max}$|GPP, T|ET, WUE and $g_s$|$G_s$ responses to
D (Figure 5). The difference in magnitude of GPP and $A_{max}$ at high D may be explained by the
proportion of shaded leaves: LAI is around 1 $m^2$ [leaf area] $m^{-2}$ [ground area], a proportion of these
leaves are in the shadow, it is thus expected that $A_{max}$ (expressed per $m^2$ of leaves) will be higher. The
similar magnitude for $G_s$ and $g_s$ was also expected, as LAI is close to 1 and $R_n$ is not a driver for
stomatal conductance. The peaked pattern of T versus D, as opposite to plateaued pattern of ET, may
be explained by (1) the contribution of soil evaporation to ET or (2) the presence of mistletoe, known
for not regulating their stomata (Griebel et al., 2017). The higher magnitude of leaf water use
efficiency results from the combination of higher $A_{max}$ and similar or lower leaf transpiration
compared to ET.





Our study demonstrated that canopy dynamics play an important role in regulating seasonal variations
in GPP even in evergreen forests. Similar observations emerged from a tropical forest, where leaf area
index and leaf age explained the seasonal variability of GPP (Wilson et al., 2001;Wu et al., 2016), as
the photosynthetic capacity (PC, the maximum rate of GPP in optimal environmental condition)
varied with leaf age. In the Cumberland Plain forest, periods with high LAI co-occur with mature,
efficient leaves, and periods with low LAI co-occur with old, less efficient leaves. LAI was correlated
with PC, which was probably the result of both a greater number of leaves and more efficient leaves.
Remotely sensed vegetation indices such as enhanced vegetation index (EVI) or normalized
difference vegetation index (NDVI) assess whether the target being observed contains live green
vegetation. In Australia, NDVI and EVI were good predictors of photosynthetic capacity in savanna,
mulga and Mediterranean-mallee ecosystems (Restrepo-Coupe et al., 2016). However, for
Cumberland Plain forest NDVI was a poor predictor of PC, which could be explained because (1)
greenness did not drive photosynthetic potential, which could be because the understory greening has
only subtle influence on PC or (2) there is a bias or a scale mismatch in the NDVI measurements.
Unfortunately, satellite-derived LAI values are typically inaccurate in open forests and forests in
southeast Australia (Hill et al., 2006), which might also indicate limitations of satellite products to
establish successful relations between NDVI and GPP in sclerophyll ecosystems.
In a global study, it was shown that mean annual NEE decreased with increasing dryness index
(PET/P) in sites located below 45° N latitude (Yi et al., 2010). It has also been shown that *Eucalyptus*
grow more slowly in warm environments (Prior and Bowman, 2014). At Cumberland Plain, and in a
previous study (van Gorsel et al., 2013), GPP decreased with D above a threshold of ~ 1.3 kPa. Our
results indicate that surface conductance ($G_s$) decreased above that threshold, suggesting that the
decrease in GPP is caused by stomatal regulation. As D correlates with air temperature, it is difficult
to distinguish the relative contribution of D and $T_a$ to the decrease of $G_s$, but they are thought to both
impact $G_s$ (Duursma et al., 2014). The Cumberland Plain has the highest mean annual temperature and
the highest dryness index among the four Eucalyptus forest eddy-covariance sites in south-east
Australia (Beringer et al. 2016), which could explain its unique seasonality.





## 5. Conclusions

The Cumberland Plain forest was a net C source in summer and a net C sink in winter, in contrast to other Australian eucalypt forests which were net C sinks year-round. ER drove NEE seasonality, as the seasonal amplitude of ER was greater than GPP. ER was high in the warmer, wetter months of summer, when environmental conditions supported high autotrophic respiration and heterotrophic decomposition. Meanwhile, GPP was limited by lower LAI and older leaves in early summer, and by high D which limited $G_s$ throughout the summer. Despite being evergreen, there was significant temporal variation in LAI, which was correlated with monthly photosynthetic capacity. Understanding LAI dynamics and its response to precipitation regimes will play a key-role in climate change feedback.

## Code and data availability

All the datasets and scripts used in this manuscript can be downloaded at:
https://doi.org/10.5281/zenodo.1069862

## Author contribution

DT, VRD, EP, AAR conceived the project; CVMB, CM, EP, AAR, AG, MMB, collected the data and assured the maintenance of the experiment; AAR, AG, CAW, EP, PI, VRD, analysed the data; AAR, EP, VDR wrote the manuscript with input from all other authors.

The authors declare that they have no conflict of interest.

## Acknowledgements

The Australian Education Investment Fund, Australian Terrestrial Ecosystem Research Network, and Hawkesbury Institute for the Environment at Western Sydney University supported this work. We thank Jason Beringer, Helen Cleugh, Ray Leuning, Dan Metzen and Eva van Gorsel for advice and support. Senani Karunaratne provided soil classification details.





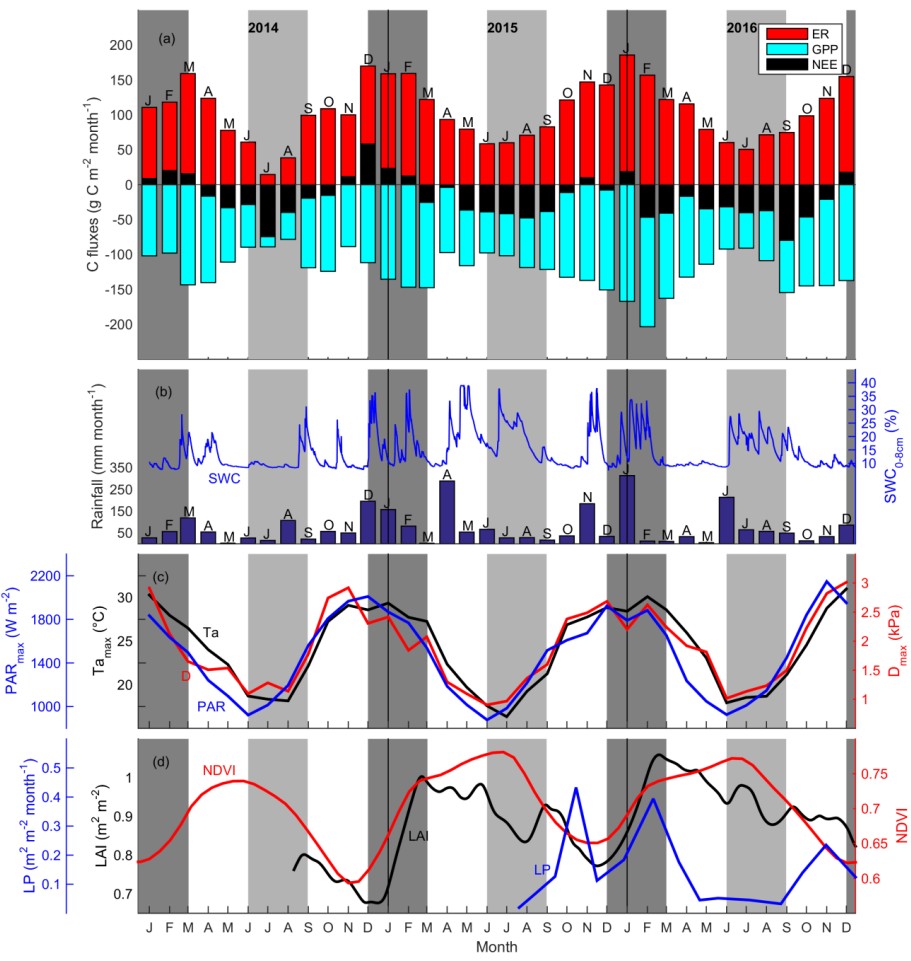

419

**Figure 1** a) Time series of monthly carbon flux (net ecosystem exchange (NEE), ecosystem respiration (ER) and gross
primary productivity (GPP), g C m$^{-2}$ month$^{-1}$) (negative indicates ecosystem uptake); b) rainfall, mm month$^{-1}$; soil water
content from 0 to 8 cm (SWC 0-8cm, %); c) average of daily maximum for each month photosynthetically active radiation
(PARmax, W m$^{-2}$), air temperature (Ta$_{max}$ , ℃) and vapour pressure deficit (D$_{max}$, kPa)]. Canopy dynamics trends
[normalized difference vegetation index (NDVI, unitless); d) leaf area index (LAI, m$^2$ m$^{-2}$) from November 2013 to April
2016 and litter production (LP, m$^2$ m$^{-2}$ month$^{-1}$)]. Shaded areas shows summer (dark grey) and winter (light grey). Note
Ta$_{max}$ and PAR$_{max}$ remained above 15 ℃ and 800 W m$^{-2}$.







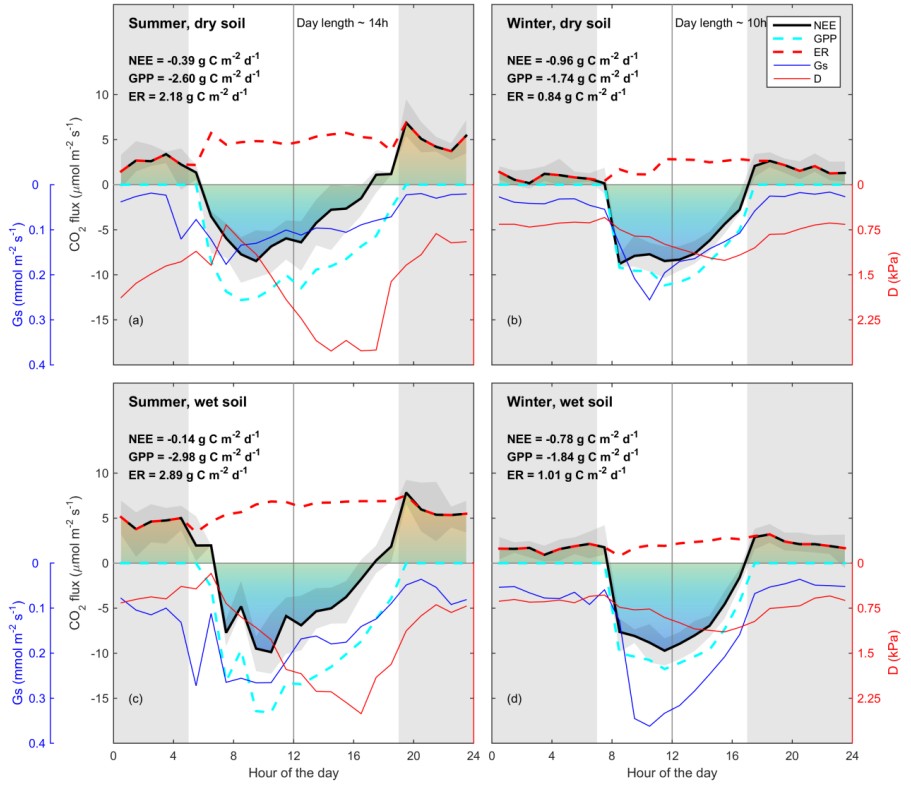

**Figure 2** Diurnal trend (line: median and shade: quartile) of clear-sky measured net ecosystem exchange (NEE, thick black
line, $\mu$mol m$^{-2}$ s$^{-1}$); estimated daytime ecosystem respiration (ER, inferred from a neural network fitted on nighttime NEE,
thick dotted red line, $\mu$mol m$^{-2}$ s$^{-1}$); estimated gross primary productivity (GPP, inferred as NEE – estimated daytime ER,
thick dotted cyan line, $\mu$mol m$^{-2}$ s$^{-1}$); measured vapour pressure deficit (D, thin red line, kPa); and estimated surface
conductance (G$_s$, inferred from Penman-Monteith, blue line, mmol m$^{-2}$ s$^{-1}$). Grey shade shows night-time (sunset to sunrise).
NEE, GPP and ER number are calculated by integrating the diurnal fluxes as shown in the figure. "Wet" and "dry" soil is
defined as below or above the median of soil water content during summer or winter. Summer is December through
February. Winter is June through August, as defined by the Sydney bureau of meteorology. Colours under NEE rate are
shown for visualisation. Note that there is an asymmetry between morning and afternoon NEE in summer, but not in winter.
Note that ecosystem respiration (nighttime NEE) is enhanced by SWC in summer, but not in winter. Data used in this figure
correspond to clear-sky half-hour values, where high quality measured data for NEE were available.





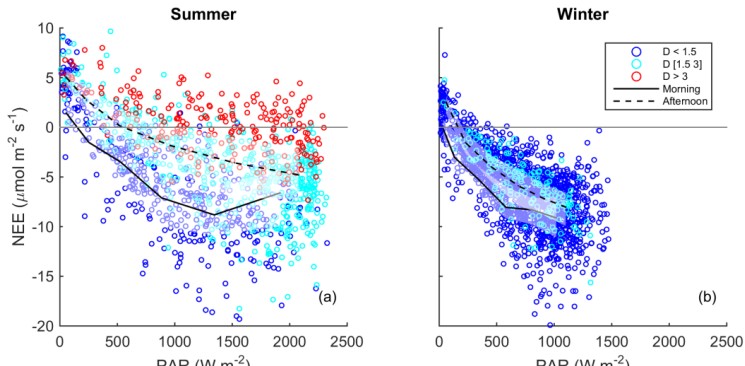

**Figure 3** Half-hourly measured NEE vs. PAR, coloured by D (blue, D < 1.5 kPa, cyan: D [1.5-3] kPa, red: D > 3 kPa) for (a) summer, and (b) winter periods. Raw data are binned by light levels to show median (lines) and quartiles (white shades) for morning (continuous lines) and afternoon (dotted lines) hours separately.





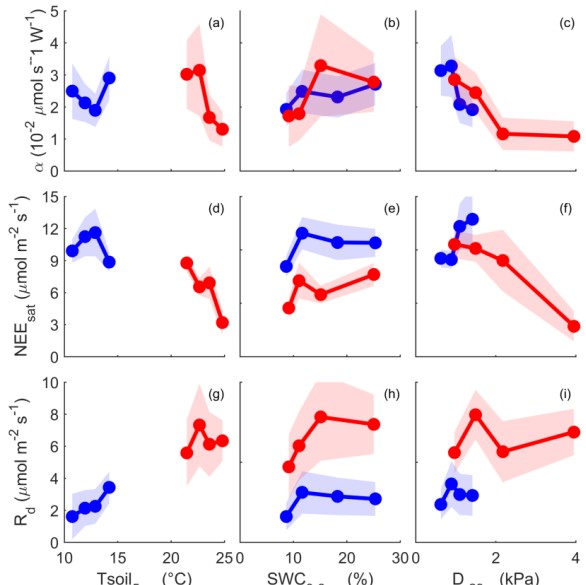

**Figure 4** NEE μmol m$^{-2}$ s$^{-1}$ light response parameters, calculated for different bins of climatic drivers (soil temperature (T$_{soil}$, °C) at 5cm depth, soil water content (SWC, %) from 0 cm to 8 cm depth, and atmospheric demand (D, kPa) at 30 m height), only raw, qc filtered daytime data is used. Light response curve was fitted using Mitscherlich equation (see methods), α is the initial slope, near PAR = 0 (μmol s$^{-1}$ W$^{-1}$), NEE$_{sat}$ μmol m$^{-2}$ s$^{-1}$ is NEE at light saturation, and R$_d$ μmol m$^{-2}$ s$^{-1}$ is the dark respiration (NEE when PAR = 0). Blue indicates winter months, Red indicates summer months. Dots are parameters value for each quartile of driver, plotted at x = median of driver for each bin. Shading is 95% confidence interval of the parameter fit.



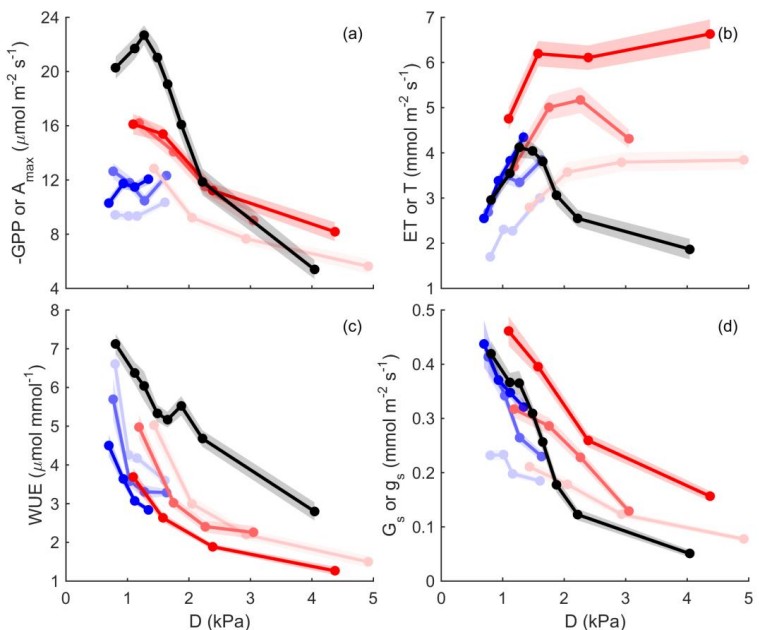

**Figure 5** Gross primary productivity or net assimilation (GPP or Amax, μmol m$^{-2}$ [ground or leaf] s$^{-1}$), evapotranspiration or leaf transpiration (ET or T, mmol m$^{-2}$ [ground or leaf] s$^{-1}$), water use efficiency (WUE = GPP/ET or A$_{max}$/T, μmol mmol$^{-1}$) and surface conductance or leaf conductance (G$_s$ or g$_s$, mmol m$^{-2}$ s$^{-1}$) vs. vapour pressure deficit (D). Leaf level is shown in black, ecosystem scale is shown in color; summer (red) and winter (blue), at saturated PAR (>1000 W m$^{-2}$). D is binned into 4 quartiles for ecosystem and 8 for leaf; Y is mean value for each D bins, plotted at the median of D bin. Shaded area indicates the standard error of the mean. The three color intensity show SWC quantiles (SWC < 0.33, SWC [0.33-0.67] and SWC [0.67-1.00] shown in decreasing color intensity).



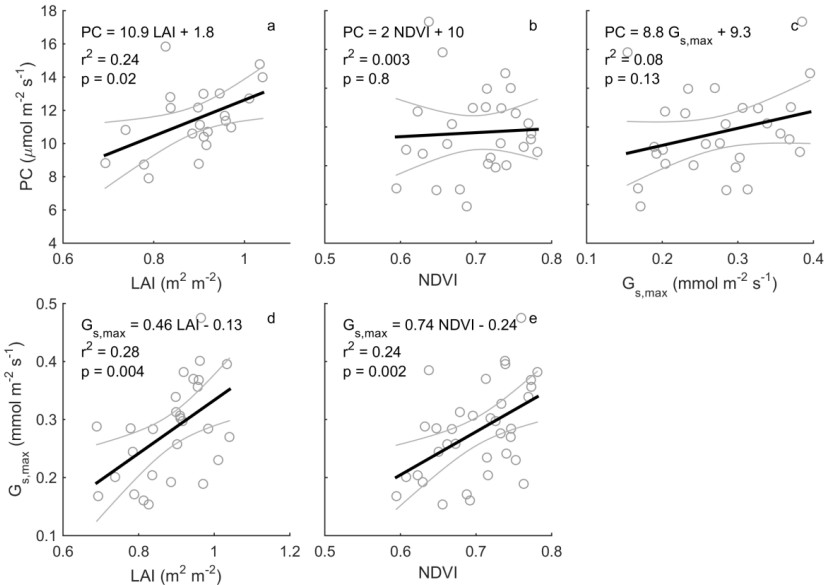

465

**Figure 6** Relationships between monthly photosynthetic capacity (PC, µmol m$^{-2}$ s$^{-1}$), leaf area index (LAI, m$^2$ m$^{-2}$), 250 m$^2$ normalized vegetation index (NDVI), and maximum surface conductance (G$_{s,max}$). Monthly PC / G$_{s,max}$ are calculated as the median / 75% quantile of half-hourly GPP / G$_s$ when PAR [800-1200 W m$^{-2}$] and D [1-1.5 kPa]; rain events are filtered for G$_{s,max}$ estimation, to minimise evaporation contribution to evapotranspiration (see methods). Monthly LAI is calculated as mean of LAI smoothed by a spline. Thick black line shows a linear regression. PC vs. LAI was significant (r$^2$ = 0.58, p = 3.7e-05, slope = 20.1 ± 3.8). G$_{s,max}$ vs. LAI was also significant (r$^2$ = 0.28, p = 0.004, slope = 0.46 ± 0.15). For PC calculation, GPP data is only used when quality-checked NEE is available (GPP = NEE measured – ER estimated by a neural network, see method).




**Table 1** Annual precipitation (P, mm y⁻¹), evapotranspiration (ET, mm y⁻¹), air temperature $T_a$ (°C), net ecosystem exchange
(NEE, g C m⁻² y⁻¹), gross ecosystem production (GPP, g C m⁻² y⁻¹) and ecosystem respiration (ER, g C m⁻² y⁻¹) for the three
year study period.

| Period | P (mm y⁻¹) | ET (mm y⁻¹) | $T_a$ (°C) | NEE (g C m⁻² y⁻¹) | GPP (g C m⁻² y⁻¹) | ER (g C m⁻² y⁻¹) |
|---|---|---|---|---|---|---|
| **2014 all** | **733** | **802** | **18.4** | **-54** | **-1347** | **1293** |
| winter | 149 | 142 | 12.8 | -127 | -268 | 141 |
| spring | 129 | 193 | 19.3 | -8 | -355 | 347 |
| summer | 279 | 275 | 22.8 | 97 | -338 | 434 |
| autumn | 176 | 192 | 18.8 | -15 | -385 | 370 |
| **2015 all** | **978** | **935** | **18.0** | **-183** | **-1641** | **1458** |
| winter | 122 | 160 | 12.0 | -135 | -322 | 187 |
| spring | 237 | 223 | 19.1 | -28 | -429 | 401 |
| summer | 273 | 318 | 23.0 | 31 | -481 | 512 |
| autumn | 345 | 234 | 18.1 | -51 | -410 | 359 |
| **2016 all** | **893** | **849** | **18.8** | **-337** | **-1735** | **1398** |
| winter | 335 | 161 | 13.1 | -99 | -283 | 185 |
| spring | 96 | 208 | 18.8 | -135 | -469 | 334 |
| summer | 412 | 308 | 23.7 | -15 | -560 | 545 |
| autumn | 50 | 172 | 19.6 | -88 | -422 | 334 |





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
