# Peer review of "Upside-down fluxes Down Under: CO2 net sink in winter and"

_Biogeosciences, 2017_

## Referee Comment (RC1) · Anonymous Referee #1 · 14 Feb 2018

General comments: Renchon et al. present a well-written analysis of the net ecosystem exchange of an evergreen sclerophyll forest near Sydney. Using eddy covariance techniques, the authors demonstrate that the forest behaves in contrast to the conventional pattern where hot, rainy summers are more productive than mild, dry winters. Instead, the higher microbial respiration in the wet summer season outweighs the benefit of increased summer photoperiod causing the forest to be a net carbon source in the summer and sink in the winter. The authors use additional vegetation/leaf level measurements of photosynthesis and conductance to strengthen their assertions of flux dynamics, although the discussion of these results feels rather like a tangential addition. These findings could be more strongly incorporated into the discussion of dif-

ferences in fluxes from old and young leaves and into the discussion of the influence of diurnal cycles and hysteresis on seasonal trends. The work of A. Griebel on anisohydricity induced by mistletoes in these ecosystems could be drawn upon in greater detail during the discussion of leaf specific measurements and trends in seasonal response to moisture availability as it is likely as highly relevant to ecosystem carbon dynamics as it is to water dynamics.

Specific concerns: The authors conclude that diurnal patterns of NEE, GPP, and ER have central roles in determining the seasonal carbon source/sink dynamics, but a stronger analysis would significantly bolster this claim –a wavelet coherence analysis of the time series could be an informative addition that would support this conclusion more thoroughly. A more complete discussion of the old and young leaf-level data would likewise improve support for the conclusion that GPP was limited by leaf age in the summer. Following these additions, strong conclusions regarding the influence of global climate change on the future carbon exchange in these forests can be drawn. Lastly, while this paper makes a significant contribution without it, an analysis of GPP in comparison to solar induced fluorescence (SIF) which has been recently shown as a better proxy for GPP than NDVI may prove very interesting.

Technical corrections: L243: Were leaves actually measured at 1.5km height as this sentence suggests? From what light environment were measured leaves collected?

---

## Referee Comment (RC2) · Anonymous Referee #2 · 17 Feb 2018

This manuscript describes three years of ecosystem carbon (C) and water flux observations in a relatively dry temperate evergreen forest. Interesting temporal dynamics of C fluxes are described whereby uptake was highest during the winter months and largely driven by variations in ecosystem respiration (ER). Analyses of controls on fluxes are presented and demonstrate the important role of elevated air temperatures and vapor pressure deficits in restricting surface conductance and limiting gross primary production (GPP) during summer months. Although this was an evergreen forest, phenology and structural dynamics of the canopy were important determinants of photosynthesis.

This manuscript presents material that would be appropriate for the readership of Biogeosciences, provided that the points of concern and comments outlined below can be adequately addressed.

**Comments**

Please eliminate the use of the diffusely defined term "atmospheric drought". A term like "atmospheric demand" is more precise and appropriate.

Is it really true that the canopy height is ~25 m (L105) and the top of the profile and eddy covariance system is 29 m? This is rather close to a tall canopy for an eddy covariance application for observing ecosystem fluxes. A check of the site description is needed, and if indeed these numbers correct, a clearer presentation and discussion of the implications on representativeness of reported fluxes and analyses is in order.

Was NDVI measured at the site or was a satellite product used? What was the rationale for using only NDVI versus EVI (or checking both)?

A careful read and editing of the methods (and entire manuscript) is needed to ensure better consistency in the use of terminology and symbols be used. Furthermore, where possible, the use of more common symbols/abbreviations would be helpful. A non-exhaustive list of examples includes:

- $F_{CT}$ and $F_{CS}$ are used to represent the eddy flux and storage flux, respectively (Eq. 1). Then Fc and Sc are used to represent the eddy flux and storage flux, respectively (L136).
- $F_N$ (e.g., L183-184) and Rn (L196) are used alternately to represent net radiation.
- The profile system measured CO2 "mixing ratios" (L115), then "concentration" is used later (L160-170) with a symbol similar to the "concentrations" referred to in relation to the high frequency density measurements made by the open path IRGA.
- Use Δ or $s$ for the slope of the saturation vapor pressure curve instead of ε
- Use $LE$ or λ$E$ for latent heat flux

The sign conventions regarding the directions of fluxes are mixed up in places. For example, in the abstract C sinks carry a positive sign for uptake (L18-19), but later in the text (L264-265) "C sinks" are reported with negative signs. Please carefully review the entire manuscript and ensure consistency throughout regarding sign conventions for fluxes, sinks, and sources.

L111…The LI190SB quantum sensor is calibrated to report PPFD as umol/m$^2$/s. Was this then converted to W/m$^2$? Check the units throughout the manuscript because the reported values for incident PAR in W/m$^2$ (e.g., L230-231 and 242, Figs. 1 and 3) are not physically possible. Also check to ensure that there was no effect on analyses and it is only an error in the manuscript text.

L129-137: If the net ecosystem exchange reduces to the sum of the eddy flux and the storage flux, then don't worry about including advection in the equation. Just state the simplifying assumptions clearly in full in the text. Note that more than just well-developed turbulence (L133-134) is needed to simplify the mass balance on the control volume (e.g., horizontal homogeneity). Please be more complete in this description in the text. Were there any concerns regarding the validity of the simplifying assumptions because of proximity of the EC system to the canopy (as mentioned previously)?

On the calculation of the eddy flux:

- It is more accurate to state that the IRGA measures the (number) densities of $CO_2$ and water vapor (L142).
- Eq. 2 is not necessary with an adequate description in the text (but if you keep it define primes and the overbar). It's not the most elegant presentation in the current Eq 2…especially since the equation doesn't include the WPL terms, which are needed. Given the maturity of the EC method a text description is fine.
- L148-149. Rephrase: "Fluxes were rotated into the natural wind coordinate system using the double rotation method". Wilczak et al. isn't the best reference for the double rotation. The original is Tanner, C. B. and Thurtell, G. W.: 1969, Anemoclinometer Measurements of Reynolds Stress and Heat Transport in the Atmospheric Surface Layer, University of Wisconsin Tech. Rep., ECOM-66-G22-F, 82 pp. [Available from US Army Electronic Command, Atmospheric Sciences Laboratory, Ft. Huachuca, AZ 85613.] or referencing the chapter in the Handbook of Micrometeorology.
- L149…what time lags? Between the sonic and IRGA?
- L150-151. Block averaging is not a detrending operation.
- Check to make sure that the order of the steps in the description of the flux calculation matches what was actually done (e.g., one of the last items in the description concerns the removal of spikes in raw data, L153).

On the calculation of the storage flux:

- A complete description of Eq (4) is lacking (definition of all symbols etc.)
- Why were storage fluxes of water vapor not estimated? Is this a significant source of bias in LE measurements?

L183. What exactly is specific heat density (SHD)?

L188-194. It would be worth presenting a footprint climatology in the supplementary information.

L195-199. Clarify whether closure was forced on the fluxes reported in the results.

L200-210. Why were the turbulent fluxes not substituted for available energy when calculating surface conductance? The spatial representativeness would be better.

Eq 7. Check the 2nd term in the denominator on the RHS…the exponent should be -0.67. Add a citation and be sure all terms are defined.

L241-244. The description of leaf-level sampling needs more detail. Since there is a reference that describes these measurements in more detail, the description in this manuscript can be abbreviated, however, a more thorough description of the basics.

- The instrument used
- More details on leaf chamber conditions…were the temps, humidity etc matched to ambient?
- Were sunlit or shaded leaves (or both) measured?
- What species were targeted?

L266-267. "Summer GPP was higher (-460 ± 112 g C m-2) compared to winter GPP (-291 ± 28 g C m-2)": -291 is higher (>) than -460. Check the manuscript for any other discrepancies when comparing magnitude and direction and ensure that the wording is correct.

L272-278. It looks like there is still hysteresis during winter (albeit less severe than in summer, Figs. 2 and 3). It might be useful to add 2 panels to Fig. 3 and show surface conductance light responses to help with underscoring the importance of stomatal regulation of C fluxes.

L336-337. Is soil respiration in the subsoil really that important to the integral over the whole profile?

L345-356. The paragraph starts out by rather definitively stating that "strong stomatal regulation" was the driver of diurnal hysteresis in NEE during summer and then becomes less clear and murkier. Seems odd to take this approach if it is was found that there was strong stomatal regulation of GPP and NEE. Revise.

L357-366. Hard to follow…especially the first sentence. Revise and clarify.

L367. "Canopy dynamics"…be more specific about what you are referring to.

L390-393. Is the temperature/moisture regime at Cumberland Plain the only difference versus the other sites in eucalyptus forests? Are all the sites at similar ages and stages of succession?

Figs. 2 &. It still looks like there is hysteresis in winter…is it that it is not statistically significant?

Fig. 4. Check units for apparent quantum yields (α). These values seem high.

---

## Author Comment (AC1) · 11 Mar 2018

**Reviewer 1 comment:**

The authors use additional vegetation/leaf level measurements of photosynthesis and conductance to strengthen their assertions of flux dynamics, although the discussion of these results feels rather like a tangential addition. These findings could be more strongly incorporated into the discussion of differences in fluxes from old and young leaves and into the discussion of the influence of diurnal cycles and hysteresis on seasonal trends.

**Response:**

We will add more discussion and citations in the discussion.

**Reviewer 1 comment:**

The work of A. Griebel on anisohydricity induced by mistletoes in these ecosystems could be drawn upon in greater detail during the discussion of leaf specific measurements and trends in seasonal response to moisture availability as it is likely as highly relevant to ecosystem carbon dynamics as it is to water dynamics.

**Response:**

Thank you for this suggestion. We would like to include discussion of the effects of biotic disturbance by mistletoe at our site, but these data are not yet publicly available. Including more detailed leaf-level results is beyond the scope of this manuscript.

**Reviewer 1 comment:**

The authors conclude that diurnal patterns of NEE, GPP, and ER have central roles in determining the seasonal carbon source/sink dynamics, but a stronger analysis would significantly bolster this claim – a wavelet coherence analysis of the time series could be an informative addition that would support this conclusion more thoroughly.

**Response:**

Thank you for this suggestion. We will conduct a wavelet coherence analysis to assess the coherence between GPP and D during the manuscript revision process. We will include this figure if it provides additional insights into the processes regulating these fluxes. We need to make sure to avoid circularity of reasoning between other drivers such as temperature and NEE or ER because temperature was used as a driver in the gap-filling process.

**Reviewer 1 comment:**

A more complete discussion of the old and young leaf-level data would likewise improve support for the conclusion that GPP was limited by leaf age in the summer. Following these additions, strong conclusions regarding the influence of global climate change on the future carbon exchange in these forests can be drawn.

**Response:**

Thank you for this suggestion. We are in the process of retrieving relevant leaf-level gas exchange data that accounts for differences in leaf age. If successful, this data will allow us to make stronger inferences about the mechanisms limiting GPP at the site.

**Reviewer 1 comment:**

Lastly, while this paper makes a significant contribution without it, an analysis of GPP in comparison to solar induced fluorescence (SIF) which has been recently shown as a better proxy for GPP than NDVI may prove very interesting.

**Response:**

Thank you for this suggestion. At present, this analysis is beyond the scope of our work, but it will be considered for future analyses.

**Reviewer 1 comment:**

L243: Were leaves actually measured at 1.5km height as this sentence suggests? From what light environment were measured leaves collected?

**Response:**

Thank you for pointing out the typo, it has been corrected.

---

## Author Comment (AC2) · 11 Mar 2018

The authors thank the reviewers for thorough reading and comments to the manuscript. Please find responses to specific comments below.

Note: The authors will add a full year (2017) to the updated manuscript.

**Reviewer 2 comment:**

Please eliminate the use of the diffusely defined term "atmospheric drought". A term like "atmospheric demand" is more precise and appropriate.

**Response:**

Amended.

**Reviewer 2 comment:**

Is it really true that the canopy height is ~25 m (L105) and the top of the profile and eddy covariance system is 29 m? This is rather close to a tall canopy for an eddy covariance application for observing ecosystem fluxes. A check of the site description is needed, and if indeed these numbers correct, a clearer presentation and discussion of the implications on representativeness of reported fluxes and analyses is in order.

**Response:**

We agree that our initial statement was misleading, so we used airborne lidar data to inspect the structural canopy properties within the footprint of the flux tower and have done a preliminary analysis the cospectrum of the sonic sensible heat flux to assess any influence of the instrument height above the canopy on the turbulent transport characteristics. We will add a figure that summarizes the canopy structure as a height profile of the frequency of lidar returns to the supplements and amend the canopy description in section 2.1 as follows:

"The flux tower is in a mature dry sclerophyll forest, with 140 Mg C ha$_{-1}$ aboveground biomass and stand density of ~500 trees ha$^{-1}$. The stand hosts a large population of mistletoe (*Amyema miquelii*), which is decreasing in abundance with increasing distance to the flux tower. The canopy structure comprises three strata, and the predominant canopy tree species are *Eucalyptus moluccana* and *E. fibrosa*. While individual trees can exceed 25 m height, an airborne lidar survey from November 2015 indicates an average canopy height of ~23 m within a 300 m radius of the flux tower (supplement figure). The mid-canopy stratum (5-12 m) is dominated by *Melaleuca decora* and the understory is dominated by *Bursaria spinosa* with various shrubs, forbs, grasses and ferns present in lower abundance."

Furthermore, we will do an in-depth analysis of the cospectrum and add a quality figure that will meet publication standards of the cospectrum of the sensible heat flux to the supplement. The preliminary cospectrum analysis indicated that the majority of the ensemble-averaged hourly cospectra of the high frequency part (1-10Hz) followed the -4/3 slope, thus we did not find any indications of systematic high frequency dampening in the cospectras. Once the final analysis is concluded we will also update the implications on turbulence characteristics in section 2.4.

**Reviewer 2 comment:**

Was NDVI measured at the site or was a satellite product used? What was the rationale for using only NDVI versus EVI (or checking both)?

**Response:**

We used a LANDSAT satellite product. The authors will add an analysis with EVI in the supplements (or in the main document if it adds new insights).

**Reviewer 2 comment:**

A careful read and editing of the methods (and entire manuscript) is needed to ensure better consistency in the use of terminology and symbols be used. Furthermore, where possible, the use of more common symbols/abbreviations would be helpful. A non-exhaustive list of examples includes:

- $F_{CT}$ and $F_{CS}$ are used to represent the eddy flux and storage flux, respectively (Eq. 1). Then $F_c$ and $S_c$ are used to represent the eddy flux and storage flux, respectively (L136).
- $F_N$ (e.g., L183-184) and $R_n$ (L196) are used alternately to represent net radiation.
- The profile system measured $CO_2$ "mixing ratios" (L115), then "concentration" is used later (L160-170) with a symbol similar to the "concentrations" referred to in relation to the high frequency density measurements made by the open path IRGA.
- Use $\Delta$ or s for the slope of the saturation vapor pressure curve instead of $\varepsilon$
- Use LE or $\lambda E$ for latent heat flux

**Response:**

We have revised the manuscript and improved the consistency of terminology throughout.

- $F_c$ and $S_c$ → $F_{CT}$ and $F_{CS}$
- $F_n$ → $R_n$
- More precision added: $CO_2$ concentrations were converted to $\mu mol \ m^{-3}$ using the ideal gas law.
- Equation (6), equation (8), $\varepsilon$ → $\Delta$
- Equation (6), text edits, latent heat flux $L$ → $\lambda E$
- Equation (7), $U$ → $w_s$

**Reviewer 2 comment:**

The sign conventions regarding the directions of fluxes are mixed up in places. For example, in the abstract C sinks carry a positive sign for uptake (L18-19), but later in the text (L264-265) "C sinks" are reported with negative signs. Please carefully review the entire manuscript and ensure consistency throughout regarding sign conventions for fluxes, sinks, and sources.

**Response:**

We have consistently adopted the usual convention from micro-meteorology, that is C going out of the atmosphere is negative (uptake), and C going into the atmosphere is positive (source).

**Reviewer 2 comment:**

L111…The LI190SB quantum sensor is calibrated to report PPFD as $\mu mol/m^2/s$. Was this then converted to $W/m^2$? Check the units throughout the manuscript because the reported values for incident PAR in $W/m^2$ (e.g., L230-231 and 242, Figs. 1 and 3) are not physically possible. Also check to ensure that there was no effect on analyses and it is only an error in the manuscript text.

**Response:**

There was an error in the manuscript text; PPFD is measured in $\mu mol \ m^{-2} \ s^{-1}$. This has been corrected throughout the text, equations and figures.

**Reviewer 2 comment:**

L129-137: If the net ecosystem exchange reduces to the sum of the eddy flux and the storage flux, then don't worry about including advection in the equation. Just state the simplifying assumptions clearly in full in the text. Note that more than just well-developed turbulence (L133-134) is needed to

simplify the mass balance on the control volume (e.g., horizontal homogeneity). Please be more complete in this description in the text. Were there any concerns regarding the validity of the simplifying assumptions because of proximity of the EC system to the canopy (as mentioned previously)?

**Response:**

We deleted the advection terms in equation (1), and added more precision regarding the assumption of negligible advection (quality flag of stationarity and turbulence development test, (Foken et al. 2004)). Relating to a comment above we also added a preliminary analysis of the turbulence characteristics to section 2.4, which will be finalized before submitting the revised manuscript (see response to second comment)

**Reviewer 2 comment:**

On the calculation of the eddy flux:

- It is more accurate to state that the IRGA measures the (number) densities of $CO_2$ and water vapor (L142).
- Eq. 2 is not necessary with an adequate description in the text (but if you keep it define primes and the overbar). It's not the most elegant presentation in the current Eq 2…especially since the equation doesn't include the WPL terms, which are needed. Given the maturity of the EC method a text description is fine.
- L148-149. Rephrase: "Fluxes were rotated into the natural wind coordinate system using the double rotation method". Wilczak et al. isn't the best reference for the double rotation. The original is Tanner, C. B. and Thurtell, G. W.: 1969, Anemoclinometer Measurements of Reynolds Stress and Heat Transport in the Atmospheric Surface Layer, University of Wisconsin Tech. Rep., ECOM-66-G22-F, 82 pp. [Available from US Army Electronic Command, Atmospheric Sciences Laboratory, Ft. Huachuca, AZ 85613.] or referencing the chapter in the Handbook of Micrometeorology.
- L149…what time lags? Between the sonic and IRGA?
- L150-151. Block averaging is not a detrending operation.
- Check to make sure that the order of the steps in the description of the flux calculation matches what was actually done (e.g., one of the last items in the description concerns the removal of spikes in raw data, L153).

**Response:**

- Concentration replaced with number density
- Equation (2) deleted, the text describes the calculation details.
- Amended.
- Amended (yes, time lags between the sonic and IRGA, which are of course much smaller in open-path yet still recommended, due to the physical distance between the two instruments).
- Amended "We applied the block averaging method to calculate each half-hour average and fluctuation relative to the average, to calculate the covariance"
- Amended, 1. Raw data screening (spikes removal …), 2. If 10% data is missing, flux will be flagged NA 3. Rotation 4. Time lags 5. Block average 6. WPL, flux calculation 7. Flags applied

**Reviewer 2 comment:**

On the calculation of the storage flux:

- A complete description of Eq (4) is lacking (definition of all symbols etc.)

- Why were storage fluxes of water vapor not estimated? Is this a significant source of bias in LE measurements?

**Response:**

- Amended: added: "Where C is $CO_2$ ($\mu mol\ m^{-3}$) and t is time (s) ($\Delta C/\Delta t$ is the variation of C over 30 minutes), z is the height (m), k [1 to n = 8] represents each inlet.
- The IRGA used for canopy storage was calibrated for water vapor only starting in 2016. We calculated storage flux of water for 2016, and the contribution of the storage to the total flux of water (turbulent exchange + change in storage) was very low. Therefore we neglected water vapor storage in the current manuscript.

**Reviewer 2 comment:**

L183. What exactly is specific heat density (SHD)?

**Response:**

This was a typo, SHD stands for specific humidity deficit.

**Reviewer 2 comment:**

L188-194. It would be worth presenting a footprint climatology in the supplementary information.

**Response:**

We will add a footprint climatology to the supplementary information as suggested.

**Reviewer 2 comment:**

L195-199. Clarify whether closure was forced on the fluxes reported in the results.

**Response:**

We did not force closure on the fluxes reported, this will be clarified in the text.

**Reviewer 2 comment:**

L200-210. Why were the turbulent fluxes not substituted for available energy when calculating surface conductance? The spatial representativeness would be better.

**Response:**

Surface conductance was calculated using the turbulent flux of water vapor coming from transpiration and the net radiation (Eq. 6). We apologise that the notation was not clear in the previous version; this has been corrected.

**Reviewer 2 comment:**

Eq 7. Check the 2nd term in the denominator on the RHS…the exponent should be -0.67. Add a citation and be sure all terms are defined.

**Response:**

Amended.

**Reviewer 2 comment:**

L241-244. The description of leaf-level sampling needs more detail. Since there is a reference that describes these measurements in more detail, the description in this manuscript can be abbreviated, however, a more thorough description of the basics.

- The instrument used
- More details on leaf chamber conditions…were the temps, humidity etc matched to ambient?
- Were sunlit or shaded leaves (or both) measured?
- What species were targeted?

**Response:**

We removed this section from the methods because we used previously published data; in the results, we provide the citation to the paper as well as the doi for the data (http://dx.doi.org/10.4225/35/55b6e313444ff).

**Reviewer 2 comment:**

L266-267. "Summer GPP was higher (-460 ± 112 g C m-2) compared to winter GPP (-291 ± 28 g C m-2)": -291 is higher (>) than -460. Check the manuscript for any other discrepancies when comparing magnitude and direction and ensure that the wording is correct.

**Response:**

Amended: Summer GPP indicated greater uptake (-460 ± 112 g C m$^{-2}$) compared to winter GPP (-291 ± 28 g C m$^{-2}$) (Table 1), that is a difference of ~ 169 g C m$^{-2}$.

**Reviewer 2 comment:**

L272-278. It looks like there is still hysteresis during winter (albeit less severe than in summer, Figs. 2 and 3). It might be useful to add 2 panels to Fig. 3 and show surface conductance light responses to help with underscoring the importance of stomatal regulation of C fluxes.

**Response:**

We agree that there is still hysteresis during winter, and will add the panels to Fig. 3 indicating light response of surface conductance as suggested by the reviewer.

**Reviewer 2 comment:**

L336-337. Is soil respiration in the subsoil really that important to the integral over the whole profile?

**Response:**

This is a good point. We have measured soil respiration in a nearby site (1.5 km away), showing that soil respiration can be limited in summer when surface soil is dry. This is also seen in the flux data, as shown in figure 1.

Text edit: low soil moisture in the shallow layers sometimes limited decomposition (January and February 2014, January 2015, see Figure 1), but more often shallow soil moisture was not limiting.

**Reviewer 2 comment:**

L345-356. The paragraph starts out by rather definitively stating that "strong stomatal regulation" was the driver of diurnal hysteresis in NEE during summer and then becomes less clear and murkier. Seems odd to take this approach if it is was found that there was strong stomatal regulation of GPP and NEE. Revise.

**Response:**

Agreed.

Paragraph entirely revised,

A morning-afternoon hysteresis of NEE response to PPFD occurred in summer, but not in winter (Figure 3). In winter, low D and moderately warm daytime air temperatures and high PPFD were sufficient to maintain high photosynthesis rates throughout most of the day (see Figure 1 for monthly average of daily maximum PPFD, D and $T_a$). In summer, two possible explanations of the diurnal hysteresis of NEE are (1) ER is greater in the afternoon compared to morning or (2) GPP is lower in the afternoon compared to morning. Explanation (1) is plausible, as temperature drives autotrophic and heterotrophic respiration; however, it is unlikely to explain the hysteresis magnitude which is much higher in summer compared to winter. Explanation (2) could arise from lower afternoon stomatal conductance or lower photosynthetic capacity (e.g. Vcmax decrease above at high $T_a$), or a combination of both or even circadian regulation (de Dios et al. 2015; Jones et al. 1998). An analysis of surface conductance showed strong stomatal regulation (Figure 2, Figure 5), induced by high atmospheric demand and high air temperature (Duursma et al. 2014), limiting photosynthesis during the afternoon of warm months (see Figure S6). These diurnal patterns of NEE, GPP and ER play a strong role in regulating the seasonal carbon cycling dynamics in this ecosystem.

**Reviewer 2 comment:**

L357-366. Hard to follow…especially the first sentence. Revise and clarify.

**Response:**

Agreed.

We revised the first sentence:

We analysed leaf-level and ecosystem level gas exchange response to D and SWC (leaf $A_{max}$ and ecosystem GPP, leaf T and ecosystem ET –filtered for rain events to minimise E, leaf WUE ($A_{max}/T$) and ecosystem WUE (GPP/ET), leaf $g_s$ and ecosystem $G_s$). We observed comparable responses (Figure 5).

**Reviewer 2 comment:**

L367. "Canopy dynamics"…be more specific about what you are referring to.

**Response:**
Right. We refer to leaf area index. Text amended.

Canopy dynamics → canopy dynamics (specifically, LAI in our analysis)

**Reviewer 2 comment:**

L390-393. Is the temperature/moisture regime at Cumberland Plain the only difference versus the other sites in eucalyptus forests? Are all the sites at similar ages and stages of succession?

**Response:**

The authors will gather more information about ages and stages of succession from other sites and add the information in the discussion.

**Reviewer 2 comment:**

Figs. 2 &. It still looks like there is hysteresis in winter…is it that it is not statistically significant?

**Response:**

There is a hysteresis in winter, but of lower magnitude. The text has been corrected accordingly.

**Reviewer 2 comment:**

Fig. 4. Check units for apparent quantum yields (α). These values seem high.

**Response:**

The units states that the y-axis is multiplied by $10^{-2}$, i.e. quantum yields value are around 0.02 (µmol $CO_2$ µmol photon$^{-1}$), values comparable to (Aubinet et al. 2001, fig. 10). Note that Mitscherlich equation (7) fits the slope to NEE, not to GPP (as in Michaelis-Menten), so the slope account for ER and are bigger.

**References:**

de Dios, V. R., A. W. Fellows, R. H. Nolan, M. M. Boer, R. A. Bradstock, F. Domingo, and M. L. Goulden, 2015: A semi-mechanistic model for predicting the moisture content of fine litter. *Agricultural and Forest Meteorology*, **203,** 64-73.
Duursma, R. A., and Coauthors, 2014: The peaked response of transpiration rate to vapour pressure deficit in field conditions can be explained by the temperature optimum of photosynthesis. *Agricultural and Forest Meteorology*, **189,** 2-10.
Foken, T., M. Gockede, M. Mauder, L. Mahrt, B. Amiro, and W. Munger, 2004: Post-field data quality control. *Handbook of Micrometeorology: A Guide for Surface Flux Measurement and Analysis*, **29,** 181-208.
Jones, T. L., D. E. Tucker, and D. R. Ort, 1998: Chilling delays circadian pattern of sucrose phosphate synthase and nitrate reductase activity in tomato. *Plant Physiology*, **118,** 149-158.

---

## Author Response (AR1)

The authors thank the reviewers for thorough reading and comments to the manuscript. Please find responses to specific comments below.

Note: The authors added a full year of data (2017) to the updated manuscript.

**Reviewer 1 comment:**

The authors use additional vegetation/leaf level measurements of photosynthesis and conductance to strengthen their assertions of flux dynamics, although the discussion of these results feels rather like a tangential addition. These findings could be more strongly incorporated into the discussion of differences in fluxes from old and young leaves and into the discussion of the influence of diurnal cycles and hysteresis on seasonal trends.

[…] A more complete discussion of the old and young leaf-level data would likewise improve support for the conclusion that GPP was limited by leaf age in the summer. Following these additions, strong conclusions regarding the influence of global climate change on the future carbon exchange in these forests can be drawn.

**Response:**

Thank you for this suggestion. Although we lack site-specific data on leaf age effects on $A_{max}$, we incorporated a relevant reference to the discussion: "In Australian woodlands, PC ($A_{max}$) of leaves was also found to decrease with leaf age, $A_{max}$ declined by 30% on average between young and old leaves, for 10 different species (Reich et al., 2009).", L391-393. This suggests that both leaf age and LAI are important in limiting GPP at our site.

**Reviewer 1 comment:**

The work of A. Griebel on anisohydricity induced by mistletoes in these ecosystems could be drawn upon in greater detail during the discussion of leaf specific measurements and trends in seasonal response to moisture availability as it is likely as highly relevant to ecosystem carbon dynamics as it is to water dynamics.

**Response:**

Thank you for this suggestion. We would like to include discussion of the effects of biotic disturbance by mistletoe at our site, but these data are not yet publicly available. Moreover, detailed leaf-level responses are beyond the scope of this manuscript.

**Reviewer 1 comment:**

The authors conclude that diurnal patterns of NEE, GPP, and ER have central roles in determining the seasonal carbon source/sink dynamics, but a stronger analysis would significantly bolster this claim – a wavelet coherence analysis of the time series could be an informative addition that would support this conclusion more thoroughly.

**Response:**

Thank you for this suggestion. We conducted a wavelet coherence analysis to assess the coherence between GPP and D (figure S11). We did not add the figure to the main text, as the observed coherence between GPP and D is mostly due to daily cycle (GPP and D high during the day, low during night) and annual cycle (GPP and D high in summer, low in winter). Moreover, D covaries with other drivers such as PPFD and Ta, which makes it difficult to attribute coherence solely to D. Some interesting coherence appeared at weekly and monthly period. Furthermore, wavelet coherence is useful to quantify phase lags, but this is beyond the scope of the current manuscript.

**Reviewer 1 comment:**

Lastly, while this paper makes a significant contribution without it, an analysis of GPP in comparison to solar induced fluorescence (SIF) which has been recently shown as a better proxy for GPP than NDVI may prove very interesting.

**Response:**

Thank you for this suggestion. We note that EVI was better than NDVI for predicting PC and canopy conductance at our site. However, SIF analysis is beyond the scope of our work.

**Reviewer 1 comment:**

L243: Were leaves actually measured at 1.5km height as this sentence suggests? From what light environment were measured leaves collected?

**Response:**

Thank you for pointing out the typo. We also removed the detailed of leaf gas exchange measurement from section 2.13, L256-258, as we used previously published data.

**Reviewer 2 comment:**

Please eliminate the use of the diffusely defined term "atmospheric drought". A term like "atmospheric demand" is more precise and appropriate.

**Response:**

Amended. L30, 33, 67, 307, 341, 372 and 462

**Reviewer 2 comment:**

Is it really true that the canopy height is ~25 m (L105) and the top of the profile and eddy covariance system is 29 m? This is rather close to a tall canopy for an eddy covariance application for observing ecosystem fluxes. A check of the site description is needed, and if indeed these numbers correct, a clearer presentation and discussion of the implications on representativeness of reported fluxes and analyses is in order.

**Response:**

We agree that our initial statement was misleading, so we used airborne LiDAR data to inspect the structural canopy properties within the footprint of the flux tower and have done an analysis of the cospectra of w and $CO_2$ to assess any influence of the instrument height above the canopy on the turbulent transport characteristics (figure S3). We added a figure that summarizes the canopy structure as a height profile of the frequency of LiDAR returns to the supplements (figure S1) and amended the canopy description in section 2.1, L101-109, as follows:

"The flux tower is in a mature dry sclerophyll forest, with 140 Mg C ha$^{-1}$ aboveground biomass and stand density of ~500 trees ha$^{-1}$. The stand hosts a large population of mistletoe (*Amyema miquelii*), which is decreasing in abundance with increasing distance to the flux tower. The canopy structure comprises three strata, and the predominant canopy tree species are *Eucalyptus moluccana* and *E. fibrosa*. While individual trees can exceed 25 m height, an airborne LiDAR survey from November 2015 indicates an average canopy height of ~24 m within a 300 m radius of the flux tower (supplement figure S1). The mid-canopy stratum (5-12 m) is dominated by *Melaleuca decora* and the understory is dominated by *Bursaria spinosa* with various shrubs, forbs, grasses and ferns present in lower abundance."

Also, we added more details regarding the turbulent fluxes quality in the section 2.4, L163-165

"Although the tower height (29m) is rather close to the average canopy height (24m), cospectra analysis showed good quality turbulent fluxes (the high frequency followed the -4/3 slope, thus we did not find any indications of systematic dampening in the cospectra, see figure S3)."

**Reviewer 2 comment:**

Was NDVI measured at the site or was a satellite product used? What was the rationale for using only NDVI versus EVI (or checking both)?

**Response:**

We used LANDSAT satellite products. We replaced NDVI with EVI in the main document, as it appeared to be more closely related to canopy dynamics at the site, and moved NDVI to the supplements (see figure 1, figure 6 and figure S9, EVI follows LAI time dynamic and is better correlated with PC and $G_{s,max}$). We thank the reviewer for this useful observation.

**Reviewer 2 comment:**

A careful read and editing of the methods (and entire manuscript) is needed to ensure better consistency in the use of terminology and symbols be used. Furthermore, where possible, the use of more common symbols/abbreviations would be helpful. A non-exhaustive list of examples includes:

- $F_{CT}$ and $F_{CS}$ are used to represent the eddy flux and storage flux, respectively (Eq. 1). Then $F_c$ and $S_c$ are used to represent the eddy flux and storage flux, respectively (L136).
- $F_N$ (e.g., L183-184) and $R_n$ (L196) are used alternately to represent net radiation.
- The profile system measured $CO_2$ "mixing ratios" (L115), then "concentration" is used later (L160-170) with a symbol similar to the "concentrations" referred to in relation to the high frequency density measurements made by the open path IRGA.
- Use $\Delta$ or s for the slope of the saturation vapor pressure curve instead of $\varepsilon$
- Use LE or $\lambda E$ for latent heat flux

**Response:**

We have revised the manuscript and improved the consistency of terminology throughout.

- $F_c$ and $S_c$ $\rightarrow$ $F_{CT}$ and $F_{CS}$, L134
- $F_n$ $\rightarrow$ $R_n$, L197, L210, L218
- More precision added: $CO_2$ is measured in ppm and converted to $\mu mol\ m^{-3}$ using ideal gas law equation. L171-172
- Equation (4), equation (6), L218, $\varepsilon$ $\rightarrow$ $\Delta$
- Equation (4), L217, latent heat flux L $\rightarrow$ $\lambda E$
- Equation (5), L221, U $\rightarrow$ $w_s$

**Reviewer 2 comment:**

The sign conventions regarding the directions of fluxes are mixed up in places. For example, in the abstract C sinks carry a positive sign for uptake (L18-19), but later in the text (L264-265) "C sinks" are reported with negative signs. Please carefully review the entire manuscript and ensure consistency throughout regarding sign conventions for fluxes, sinks, and sources.

**Response:**

We have consistently adopted the usual convention from micro-meteorology, that is C going out of the atmosphere is negative (uptake), and C going into the atmosphere is positive (source). L18

**Reviewer 2 comment:**

L111…The LI190SB quantum sensor is calibrated to report PPFD as $\mu$mol/m$^2$/s. Was this then converted to W/m$^2$? Check the units throughout the manuscript because the reported values for incident PAR in W/m$^2$ (e.g., L230-231 and 242, Figs. 1 and 3) are not physically possible. Also check to ensure that there was no effect on analyses and it is only an error in the manuscript text.

**Response:**

There was an error in the manuscript text; PPFD is measured in $\mu$mol m$^{-2}$ s$^{-1}$. This has been corrected throughout the text, equations and figures. L83, 113, 235, 239, 245, 247, 252, 263, 309, 310, 320, 361, 362, 440, 442, 457, 464, 465, 472 and 479, figure 1, 3, 4 and 5, and equation (7)

**Reviewer 2 comment:**

L129-137: If the net ecosystem exchange reduces to the sum of the eddy flux and the storage flux, then don't worry about including advection in the equation. Just state the simplifying assumptions clearly in full in the text. Note that more than just well-developed turbulence (L133-134) is needed to simplify the mass balance on the control volume (e.g., horizontal homogeneity). Please be more complete in this description in the text. Were there any concerns regarding the validity of the simplifying assumptions because of proximity of the EC system to the canopy (as mentioned previously)?

**Response:**

We deleted the advection terms in equation (1), and added more precision regarding the assumption of negligible advection (quality flag of stationarity and turbulence development test, (Foken et al. 2004)). Relating to a comment above we also added a preliminary analysis of the turbulence characteristics to section 2.4, which will be finalized before submitting the revised manuscript (see response to second comment). Changes: L134-140

**Reviewer 2 comment:**

On the calculation of the eddy flux:

- It is more accurate to state that the IRGA measures the (number) densities of $CO_2$ and water vapor (L142).
- Eq. 2 is not necessary with an adequate description in the text (but if you keep it define primes and the overbar). It's not the most elegant presentation in the current Eq 2…especially since the equation doesn't include the WPL terms, which are needed. Given the maturity of the EC method a text description is fine.
- L148-149. Rephrase: "Fluxes were rotated into the natural wind coordinate system using the double rotation method". Wilczak et al. isn't the best reference for the double rotation. The original is Tanner, C. B. and Thurtell, G. W.: 1969, Anemoclinometer Measurements of Reynolds Stress and Heat Transport in the Atmospheric Surface Layer, University of Wisconsin Tech. Rep., ECOM-66-G22-F, 82 pp. [Available from US Army Electronic Command, Atmospheric Sciences Laboratory, Ft. Huachuca, AZ 85613.] or referencing the chapter in the Handbook of Micrometeorology.
- L149…what time lags? Between the sonic and IRGA?
- L150-151. Block averaging is not a detrending operation.
- Check to make sure that the order of the steps in the description of the flux calculation matches what was actually done (e.g., one of the last items in the description concerns the removal of spikes in raw data, L153).

**Response:**

L144-166 (section 2.4)

- Concentration replaced with number density, L145
- Equation (2) deleted, the section 2.4, L144-165 describes the calculation details.
- Amended.
- Amended (yes, time lags between the sonic and IRGA, which are of course much smaller in open-path yet still recommended, due to the physical distance between the two instruments).
- Amended "We applied the block averaging method to calculate each half-hour average and fluctuation relative to the average, to calculate the covariance" L156-157
- Amended, 1. Raw data screening (spikes removal …), 2. If 10% data is missing, flux will be flagged NA 3. Rotation 4. Time lags 5. Block average 6. WPL, flux calculation 7. Flags applied, L150-165

**Reviewer 2 comment:**

L168-182 (section 2.5)

On the calculation of the storage flux:

- A complete description of Eq (4) is lacking (definition of all symbols etc.)
- Why were storage fluxes of water vapor not estimated? Is this a significant source of bias in LE measurements?

**Response:**

- Amended: added: "Where C is $CO_2$ ($\mu mol\ m^{-3}$) and t is time (s) ($\Delta C/\Delta t$ is the variation of C over 30 minutes), z is the height (m), k [1 to n = 8] represents each inlet. L175-176
- The main reason not to use the $H_2O$ signal from the LI840 is that we do not have heated lines and the tubes will have significant adsorption/desorption issues depending on conditions. So the signal would not be reliable.
  We had a look at the 1-point estimation of $H_2O$ storage (from the Li-7500A at 29m), and it showed a low contribution of change in $H_2O$ storage to the total $H_2O$ flux (see figure below). Rather than using a low quality, noisy estimate of $H_2O$ from the 1-point estimation, we chose to assume that change in $H_2O$ storage was negligible.

[Figure]

**Figure 1** Diurnal course of H2o flux ($mmol\ m^{-2}\ s^{-1}$) turbulent exchange and change in storage (1 point estimation) in 2017.

**Reviewer 2 comment:**

L183. What exactly is specific heat density (SHD)?

**Response:**

This was a typo, SHD stands for specific humidity deficit. The section 2.6, L183-204, about gap-filling and partitioning, has been updated entirely with the most recent processing of the data, including 2017 for this revised manuscript.

**Reviewer 2 comment:**

L188-194. It would be worth presenting a footprint climatology in the supplementary information.

**Response:**

We added a footprint climatology to the supplementary information as suggested. (figure S5), referred to on L207 (section 2.7).

**Reviewer 2 comment:**

L195-199. Clarify whether closure was forced on the fluxes reported in the results.

**Response:**

We did not force closure on the fluxes reported, this was clarified in the text. L213-214.

**Reviewer 2 comment:**

L200-210. Why were the turbulent fluxes not substituted for available energy when calculating surface conductance? The spatial representativeness would be better.

**Response:**

Surface conductance was calculated using the turbulent flux of water vapor coming from evapo-transpiration and the net radiation (equation (4)). We apologise that the notation was not clear in the previous version; this has been corrected. L217-221, section 2.9

**Reviewer 2 comment:**

Eq 7. Check the 2nd term in the denominator on the RHS…the exponent should be -0.67. Add a citation and be sure all terms are defined.

**Response:**

Amended. Equation (5)

**Reviewer 2 comment:**

L241-244. The description of leaf-level sampling needs more detail. Since there is a reference that describes these measurements in more detail, the description in this manuscript can be abbreviated, however, a more thorough description of the basics.

- The instrument used
- More details on leaf chamber conditions…were the temps, humidity etc matched to ambient?
- Were sunlit or shaded leaves (or both) measured?
- What species were targeted?

**Response:**

We removed the details of the methods because we used previously published data; we provide the citation to the paper L258.

**Reviewer 2 comment:**

L266-267. "Summer GPP was higher (-460 ± 112 g C m-2) compared to winter GPP (-291 ± 28 g C m-2)": -291 is higher (>) than -460. Check the manuscript for any other discrepancies when comparing magnitude and direction and ensure that the wording is correct.

**Response:**

Amended: Summer GPP was lower – i.e. more uptake (-400 ± 97 g C m$^{-2}$) compared to winter GPP (-282 ± 41 g C m$^{-2}$) (Table 1), that is a difference of ~ 118 g C m$^{-2}$. L281

**Reviewer 2 comment:**

L272-278. It looks like there is still hysteresis during winter (albeit less severe than in summer, Figs. 2 and 3). It might be useful to add 2 panels to Fig. 3 and show surface conductance light responses to help with underscoring the importance of stomatal regulation of C fluxes.

**Response:**

We agree that there is still hysteresis during winter, and we adjusted the text, L291-292. Also, we added the panels to Fig. 3 indicating light response of surface conductance as suggested by the reviewer.

**Reviewer 2 comment:**

L336-337. Is soil respiration in the subsoil really that important to the integral over the whole profile?

**Response:**

This is a good point. We have measured soil respiration in a nearby site, showing that soil respiration can be limited in summer when surface soil is dry. This is also seen in the flux data, as shown in figure 1.

Text edit: low soil moisture in the shallow layers sometimes limited decomposition (January and February 2014, January and December 2015, January, February and December 2017 see Figure 1), but often regular rainfall maintained adequate soil moisture. L352-353

**Reviewer 2 comment:**

L345-356. The paragraph starts out by rather definitively stating that "strong stomatal regulation" was the driver of diurnal hysteresis in NEE during summer and then becomes less clear and murkier. Seems odd to take this approach if it is was found that there was strong stomatal regulation of GPP and NEE. Revise.

**Response:**

The authors agreed and revised the paragraph entirely (361-376):

"A strong morning-afternoon hysteresis of NEE response to PPFD occurred in summer, and less so in winter (Figure 3). In winter, low D and moderately warm daytime air temperatures and high PPFD were sufficient to maintain high photosynthesis rates throughout most of the day (Figure 1). In summer, two possible explanations of the diurnal hysteresis of NEE are (1) ER is greater in the afternoon compared to morning or (2) GPP is lower in the afternoon compared to morning. Explanation (1) is plausible, as temperature drives autotrophic and heterotrophic respiration; however, it is unlikely to explain the hysteresis magnitude which is higher in summer compared to winter. Explanation (2) could arise from lower afternoon stomatal conductance or lower photosynthetic capacity (e.g. the maximum rate of carboxylation (Vcmax) decreases at high Ta), or a combination of both or even circadian regulation (Jones et al. 1998; Resco de Dios et al. 2015). An analysis of surface conductance showed strong stomatal regulation (Figure 2, Figure 3, Figure 5), induced by high atmospheric demand and high air temperature (Duursma et al. 2014), limiting photosynthesis during the afternoon of warm months (see Figure S10). These diurnal patterns of NEE, GPP and ER play a strong role in regulating the seasonal carbon cycling dynamics in this ecosystem. A wavelet coherence analysis between D and GPP showed strong coherence at seasonal time scale (periods of three months), see figure S11."

**Reviewer 2 comment:**

L357-366. Hard to follow…especially the first sentence. Revise and clarify.

**Response:**

The authors agreed and revised the first sentence, L377-378:

"We observed comparable responses of leaf-level and ecosystem-level gas exchange to environmental drivers (Figure 5)."

**Reviewer 2 comment:**

L367. "Canopy dynamics"…be more specific about what you are referring to.

**Response:**
Right. We refer to leaf area index. Text amended.

Canopy dynamics → canopy dynamics (specifically, LAI in our analysis), L387

**Reviewer 2 comment:**

L390-393. Is the temperature/moisture regime at Cumberland Plain the only difference versus the other sites in eucalyptus forests? Are all the sites at similar ages and stages of succession?

**Response:**

The context of this statement is related to the climatic sensitivity of carbon fluxes and canopy conductance. While stand age or successional stage may affect the climatic sensitivity of the fluxes or conductance, all eucalyptus forests in the OzFlux network are mature and have been managed or disturbed to a similar degree (Hinko-Najera et al. 2017; Keith et al. 2012). The Cumberland Plain forest has somewhat lower LAI than the other forests, and has larger seasonal dynamics of LAI and carbon fluxes than Wombat Forest (Griebel et al. 2015). We have shown that GPP and conductance are strongly related to vapor pressure deficit (Fig 5) as well as LAI (Fig 6). Hence, a combination of stomatal conductance and leaf area are responsible for the climatic sensitivity of the site. We prefer not to expand the discussion on these points in order to maintain the focus of the paragraph, but we added a short phrase to link the statement to D, which we feel is the unique feature of our site (L409-412).

"Cumberland Plain has the highest mean annual temperature and the highest dryness index among the four Eucalyptus forest eddy-covariance sites in south-east Australia (Beringer et al. 2016), which could explain its strong sensitivity to D and hence its unique seasonality."

**Reviewer 2 comment:**

Figs. 2 & 3. It still looks like there is hysteresis in winter…is it that it is not statistically significant?

**Response:**

There is a hysteresis in winter, but of lower magnitude. The text has been corrected accordingly. L291-292

**Reviewer 2 comment:**

Fig. 4. Check units for apparent quantum yields (α). These values seem high.

**Response:**

In the figure 4 y-label, the units specify that the y-axis is multiplied by $10^{-2}$, i.e. quantum yields value are around 0.02 ($\mu$mol $CO_2$ $\mu$mol photon$^{-1}$), values comparable to (Aubinet et al. 2001, fig. 10). Note that Mitscherlich equation (7) fits the slope to NEE, not to GPP (as in Michaelis-Menten), so the slope accounts for ER and is bigger.

**References:**

[revised manuscript text omitted]

---

## Author Response (AR2)

The authors thank the reviewers for their comments of the manuscript. Please find responses to specific comments below.

**Reviewer comment:**

Add details in the methods section regarding the satellite data products (i.e., EVI, NDVI) used. Currently it is impossible to tell which versions were used.

**Response:**

This is a good point. We added a section in the methods section (section 2.14, L253-259)

*2.14 Remotely sensed land surface greenness*
Normalized difference vegetation index (NDVI) and Enhanced Vegetation Index (EVI) values were derived from the MODIS Terra Vegetation Indices 16-Day L3 Global 250m product (MOD13Q1), which uses atmospherically corrected surface reflectance masked for water, clouds, heavy aerosols, and cloud shadows (Didan 2015). At 250m spatial resolution, the pixel containing Cumberland Plain was assumed to be representative for the footprint and values of that pixel between 1.1.2014 and 31.12.2017 were extracted.

**Reviewer comment:**

L221. The wind speed in Eq 5 is the mean horizontal wind speed. Use a more appropriate symbol (e.g., u) that is consistent with the literature.

**Response:**

We replaced $W_s$ by U, we chose U (upper case) instead of u to avoid confusion with u*. We updated equation (5) and Line 221 "mean horizontal wind speed (U, m s$^{-1}$)

**Reviewer comment:**

L309. Here "canopy conductance" is used, but elsewhere "surface conductance" is used. Check the entire manuscript and be consistent with terminology throughout.

**Response:**

Amended. We checked throughout the text, and L309 was the only occurrence of "canopy conductance". We replaced it with "surface conductance".

**Reviewer comment:**

L477. What is "250 m2 enhanced vegetation index"?

**Response:**

We deleted the "250 m$^2$" words. The EVI methods is explained in the method section (2.14, L253-259).

**Reviewer comment:**

L478 & 479. Sentence referring to PC/Gs,max, median/75% quantile, GPP/Gs is confusing. The ratios don't appear in the figure.

**Response:**

We replaced this text with "Monthly PC and monthly $G_{s,max}$ are calculated as the median of half-hourly GPP and half-hourly $G_s$ when PPFD [800-1200 $\mu$mol m$^{-2}$ s$^{-1}$] and D [1-1.5 kPa] […]", L478-483